# Online Nonstochastic Model-Free Reinforcement Learning

**Udaya Ghai** [†][*]
Amazon
ughai@amazon.com

**Arushi Gupta** [†]
Princeton University & Google DeepMind
arushig@princeton.edu

**Wenhan Xia**
Princeton University & Google DeepMind
wxia@princeton.edu

**Karan Singh**
Carnegie Mellon University
karansingh@cmu.edu

**Elad Hazan**
Princeton University & Google DeepMind
ehazan@princeton.edu

## Abstract

We investigate robust model-free reinforcement learning algorithms designed for environments that may be dynamic or even adversarial. Traditional state-based policies often struggle to accommodate the challenges imposed by the presence of unmodeled disturbances in such settings. Moreover, optimizing linear state-based policies pose an obstacle for efficient optimization, leading to nonconvex objectives, even in benign environments like linear dynamical systems.

Drawing inspiration from recent advancements in model-based control, we introduce a novel class of policies centered on disturbance signals. We define several categories of these signals, which we term pseudo-disturbances, and develop corresponding policy classes based on them. We provide efficient and practical algorithms for optimizing these policies.

Next, we examine the task of online adaptation of reinforcement learning agents in the face of adversarial disturbances. Our methods seamlessly integrate with any black-box model-free approach, yielding provable regret guarantees when dealing with linear dynamics. These regret guarantees unconditionally improve the best-known results for bandit linear control in having no dependence on the state-space dimension. We evaluate our method over various standard RL benchmarks and demonstrate improved robustness.

## 1   Introduction

Model-free reinforcement learning in time-varying responsive dynamical systems is a statistically and computationally challenging problem. In contrast, model based control of even unknown and changing linear dynamical systems has enjoyed recent successes. In particular, new techniques from online learning have been applied to these linear dynamical systems (LDS) within the framework of online nonstochastic control. A comprehensive survey can be found in Hazan and Singh [2022]. The key innovation in the aforementioned framework is the introduction of a new policy class called Disturbance-Action Control (DAC), which achieves a high degree of representational capacity without compromising computational efficiency. Moreover, efficient gradient-based algorithms can

---

[*]Work performed while at Princeton University and Google. † denotes equal contribution.

37th Conference on Neural Information Processing Systems (NeurIPS 2023).

be employed to obtain provable regret bounds for this approach, even in the presence of adversarial noise. Crucially, these methods rely on the notion of disturbance, defined to capture unmodeled deviations between the observed and nominal dynamics, and its availability to the learner.

This paper explores the potential of applying these disturbance-based techniques, which have proven effective in model-based control, to model-free reinforcement learning. However, it is not immediately clear how these methods can be adapted to model-free RL, as the disturbances in model-free RL are unknown to the learner.

We therefore develop the following approach to this challenge: instead of relying on a known disturbance, we create a new family of signals, which we call "Pseudo-Disturbances", and define policies that use "Pseudo-Disturbance" features to produce actions. The advantage of this approach is that it has the potential to produce more robust policies. Again inspired by model-based methods, we aim to augment existing reinforcement learning agents with a "robustness module" that serves two purposes. Firstly, it can filter out adversarial noise from the environment and improve agent performance in noisy settings. Secondly, in cases where the environment is benign and simple, such as a linear dynamical system, the augmented module will achieve a provably optimal solution. We also empirically evaluate the performance of our method on OpenAI Gym environments.

## 1.1 Our Contributions

In this work, we make the following algorithmic and methodological contributions:

- In contrast to state-based policies commonly used in RL, Section 3 defines the notion of a **disturbance-based policy**. These policies augment traditional RL approaches that rely strictly on state feedback.
- We develop **three distinct and novel methods** (Sections 3.1, 3.2, 3.3) to estimate the Pseudo-Disturbance in the model-free RL setting.
- We develop a **new algorithm**, MF-GPC (Algorithm 1), which adapts existing RL methods to take advantage of our Pseudo-Disturbance framework.
- We **empirically evaluate** our method on OpenAI Gym environments in Section 5. We find that our adaptation applied on top of a DDPG baseline performs better than the baseline, significantly so in same cases, and has better robustness characteristics.
- We prove that the proposed algorithm achieves **sublinear regret** for linear dynamics in Theorem 4. These regret bounds improve upon the best-known for bandit linear control in terms of their dependence on state space dimension (Appendix E). Notably, our bounds have **no dependence on the state dimension**, reducing the state-of-the-art regret bound by factors of $\sqrt{d_x}$ for convex losses and $d_x^{2/3}$ if losses are additionally smooth, signalling that our methodology is better suited to challenging high-dimensional under-actuated settings.

## 1.2 Pseudo-Disturbance based RL

A fundamental primitive of the non-stochastic control framework is the *disturbance*. In our RL setting, the system evolves according to the following equation

$$\mathbf{x}_{t+1} = f(\mathbf{x}_t, \mathbf{u}_t) + \mathbf{w}_t \,,$$

where $\mathbf{x}_t$ is the state, $\mathbf{u}_t$ is control signal, and $\mathbf{w}_t$ is a bounded, potentially adversarially chosen, disturbance. Using knowledge of the dynamics, $f$, non-stochastic control algorithms first compute $\mathbf{w}_t$, and then compute actions via DAC, as follows

$$\mathbf{u}_t = \pi_{\text{base}}(\mathbf{x}_t) + \sum_{i=1}^{h} M_i^t \mathbf{w}_{t-i} \,.$$

Here $\pi_{\text{base}}$ is a baseline linear controller, and $M^t$ are matrices, learned via gradient descent or similar algorithms. For linear systems, the DAC law is a convex relaxation of linear policies, which allows us to prove regret bounds against powerful policy classes using tools from online convex optimization.

To generalize this approach, without a model or knowledge of the dynamics function $f$, both defining and obtaining this disturbance in order to implement DAC or similar policies becomes unclear. To

address this, we introduce the concept of a *Pseudo-Disturbance* (PD) and provide three distinct variants, each representing a novel signal in reinforcement learning. These signals have various advantages and disadvantages depending on the available environment:

1. The first notion is based on the gradient of the temporal-difference error. It assumes the availability of a value function oracle that can be evaluated or estimated online or offline using any known methodology.

2. The second notion also assumes the availability of a black-box value function oracle/generator. We assign artificial costs over the states and generate multiple auxiliary value functions to create a "value vector." The Pseudo-Disturbance is defined as the difference between the value vector at consecutive states. This signal's advantage is that it does not require any zero-order optimization mechanism for estimating the value function's gradient.

3. The third notion assumes the availability of an environment simulator. The Pseudo-Disturbance is defined as the difference between the true state and the simulated state for a specific action.

For all these Pseudo-Disturbance variants, we demonstrate how to efficiently compute them (under the appropriate assumption of either a value function oracle or simulator). We provide a reduction from any RL algorithm to a PD-based robust counterpart that converts an RL algorithm into one that is also robust to adversarial noise. Specifically, in the special case of linear dynamical systems our algorithm has provable regret bounds. The formal description of our algorithm, as well as a theorem statement, are given in Section 4. For more general dynamical systems, the learning problem is provably intractable. Nonetheless, we demonstrate the efficacy of these methods empirically.

## 1.3   Related Work

**Model-free reinforcement learning.**   Reinforcement learning [Sutton and Barto, 2018] approaches are classified as model-free or model-based [Janner et al., 2019, Ha and Schmidhuber, 2018, Osband and Van Roy, 2014], dependent on if they attempt to explicitly try to learn the underlying transition dynamics an agent is subject to. While the latter is often more sample efficient [Wang et al., 2019], model-free approaches scale better in that their performance does not prematurely saturate and keeps improving with number of episodes [Duan et al., 2016]. In this paper, we focus on adaption to unknown, arbitrary disturbances for model-free reinforcement learning algorithms, which can be viewed as a tractable restriction of the challenging adversarial MDP setting [Abbasi Yadkori et al., 2013]. Model-free approaches may further be divided into policy-based [Schulman et al., 2015, 2017], value-based approaches [Mnih et al., 2013], and actor-critic approaches [Barth-Maron et al., 2018, Lillicrap et al., 2016]; the latter use a learnt value function to reduce the variance for policy optimization.

**Robust and Adaptive reinforcement learning.**   Motivated by minimax performance criterion in robust control [Zhang et al., 2021, Morimoto and Doya, 2005] introduced to a minimax variant of Q-learning to enhance of he robust of policies learnt from off-policy samples. This was later extended to more tractable formulations and structured uncertainty sets in Tessler et al. [2019], Mankowitz et al. [2019], Pinto et al. [2017], Zhang et al. [2021], Tamar et al. [2013], including introductions of model-based variants [Janner et al., 2019]. Another approach to enhance the robustness is Domain Randomization [Tobin et al., 2017, Akkaya et al., 2019, Chen et al., 2021a], wherein a model is trained in a variety of randomized environments in a simulator, and the resulting policy becomes robust enough to be applied in the real world. Similarly, adversarial training [Mandlekar et al., 2017, Vinitsky et al., 2020, Agarwal et al., 2021] has been shown to improve performance in out-of-distribution scenarios. In contrast to the previously mentioned approaches, our proposed approach only adapts the policy to observed disturbances at test time, and does not require a modification of the training procedure. This notably means that the computational cost and sample requirement of the approach matches that of vanilla RL in training, and has the benefit of leveraging recent advances in mean-reward RL, which is arguably better understood and more studied. Adaption of RL agents to new and changing environments has been similarly tackled through the lens of Meta Learning and similar approaches [Wang et al., 2016, Nagabandi et al., 2018, Pritzel et al., 2017, Agarwal et al., 2021].

**Online nonstochastic control.** The presence of arbitrary disturbances during policy execution had been for long in the fields of robust optimization and control [Zhou and Doyle, 1998]. In contrast to minimax objectives considered in robust control, online nonstochastic control algorithms (see Hazan and Singh [2022] for a survey) are designed to minimize regret against a benchmark policy class, and thus compete with the best policy from the said class determined posthoc. When the benchmark policy class is sufficiently expressive, this approach has the benefit of robustness against adversarially chosen disturbances (i.e. non-Gaussian and potentially adaptively chosen [Ghai et al., 2021]), while distinctly not sacrificing performance in the typical or average case. The first nonstochastic control algorithm with sublinear regret guarantees was proposed in Agarwal et al. [2019] for linear dynamical systems. It was subsequently extended to partially observed systems [Simchowitz et al., 2020], unknown systems [Hazan et al., 2020], multi-agent systems [Ghai et al., 2022] and the time-varying case [Minasyan et al., 2021]. The regret bound was improved to a logarithmic rate in Simchowitz [2020] for strongly convex losses. Chen et al. [2021b] extend this approach to non-linearly parameterized policy classes, like deep neural networks. Bandit versions of the nonstochastic control setting have also been studied [Gradu et al., 2020, Cassel and Koren, 2020, Sun et al., 2023] and are particularly relevant to the RL setting, which only has access to scalar rewards.

### 1.4 Paper Outline

After some basic definitions and preliminaries in Section 2, we describe the new Pseudo-Disturbance signals and how to create them in a model-free reinforcement learning environment in Section 3. In Section 4 we give a unified meta-algorithm that exploits these signals and applies them as an augmentation to any given RL agent. In Section 5 we evaluate our methods empirically.

An overview of notation can be found in Appendix A. Appendix B contains additional experimental details. Generalization of our algorithm to discrete spaces is provided in Appendix C. Proofs for Section 3 are provided in Appendix D, while the main theore is proved in Appendix E.

## 2 Setting and Preliminaries

Consider an agent adaptively choosing actions in a dynamical system with adversarial cost functions. We use notation from the control literature: $\mathbf{x}_t \in \mathbb{R}^{d_x}$ is a vector representation of the state[2] at time $t$, $\mathbf{u}_t \in \mathbb{R}^{d_u}$ is the corresponding action. Formally, the evolution of the state will follow the equations

$$\mathbf{x}_{t+1} = f(\mathbf{x}_t, \mathbf{u}_t) + \mathbf{w}_t,$$

where $\mathbf{w}_t$ is an arbitrary (even adversarial) disturbance the system is subject to at time $t$. Following this evolution, the agent suffers a cost of $c_t(\mathbf{x}_t, \mathbf{u}_t)$.

In this work, we adapt model-free reinforcement learning algorithms to this more challenging case. The (easier) typical setting for model-free methods assume, in contrast, that the disturbance $\mathbf{w}_t$ is sampled *iid* from a distribution $\mathcal{D}$, and that the cost functions $c(\mathbf{x}, \mathbf{u})$ is fixed and known. Central to the study of model-free methods are the notions of the state and state-action value functions, defined as the discounted sum of future costs acquired by starting at any state (or state-action pair) and thereafter following the policy $\pi$. For any policy $\pi$, we denote the state and state-action value functions, which are mappings from state or state/action pair to the real numbers, as

$$Q_\pi(\mathbf{x}, \mathbf{u}) = \mathbb{E}\left[\sum_{t=0}^{\infty} \gamma^t c(\mathbf{x}_t^\pi, \mathbf{u}_t^\pi) \middle| \mathbf{x}_0^\pi = \mathbf{x}, \mathbf{u}_0^\pi = \mathbf{u}\right], V_\pi(\mathbf{x}) = \mathbb{E}\left[\sum_{t=0}^{\infty} \gamma^t c(\mathbf{x}_t^\pi, \mathbf{u}_t^\pi) \middle| \mathbf{x}_0^\pi = \mathbf{x}\right],$$

where expectations are taken over random transitions in the environment and in the policy.

A special case we consider is that of linear dynamical systems. In these special instances the state involves linearly according to a linear transformation parameterized by matrices $A, B$, i.e.

$$\mathbf{x}_{t+1} = A\mathbf{x}_t + B\mathbf{u}_t + \mathbf{w}_t.$$

---

[2]Although we consider continuous state and action spaces in this section and the remainder of the main paper, we handle discrete spaces in Appendix C.

# 3 Pseudo-Disturbance Signals and Policies

In this section we describe the three different Pseudo-Disturbance (PD) signals we can record in a general reinforcement learning problem. As discussed, the motivation for this signal comes from the framework of online nonstochastic control. We consider dynamical systems with an additive misspecification or noise structure,

$$\mathbf{x}_{t+1} = f(\mathbf{x}_t, \mathbf{u}_t) + \mathbf{w}_t,$$

where the perturbation $\mathbf{w}_t$ does not depend on the state. Using perturbations rather than state allows us to avoid recursive structure that makes the optimization landscape challenging and nonconvex. As discussed, we introduce Pseudo-Disturbance signals $\hat{\mathbf{w}}_t \in \mathbb{R}^{d_w}$ in lieu of the true disturbances. We note that the PD dimensionality $d_w$ need not be the same as that of the true disturbance, $d_x$.

An important class of policies that we consider henceforth is linear in the Pseudo-Disturbance, i.e.

$$\Pi_{\text{DAC}} = \left\{ \pi(\mathbf{x}_{1:t}) = \pi_{\text{base}}(\mathbf{x}_t) + \sum_{i=1}^{h} M_i \hat{\mathbf{w}}_{t-i} \,\middle|\, M_i \in \mathbb{R}^{d_u \times d_w} \right\}.$$

Here $\Pi_{\text{DAC}}$ denotes the policy class of Disturbance-Action-Control. The fact that $\mathbf{w}_t$ does not depend on our actions allows for convex optimization of linear disturbance-action controllers in the setting of linear dynamical systems, see e.g. Hazan and Singh [2022].

We would like to capture the essence of this favorable phenomenon in the context of model free RL, but what would replace the perturbations $\mathbf{w}_t$ without a dynamics model $f$? That's the central question of this section, and we henceforth give three different proposal for this signal.

An important goal in constructing these signals is that **in the case of linear dynamical systems, it recovers the perturbation**. This will enable us to prove regret bounds in the case the environment is an LDS.

## 3.1 Pseudo-Disturbance Class I: Value-Function Gradients

The first signal we consider is based on the gradient of the value function. The value function maps the state onto a scalar, and this information is insufficient to recreate the perturbation even if the underlying environment is a linear dynamical system. To exact a richer signal, we thus consider the gradient of the value function with respect to the action and state. The basic goal is to implement the following equation

$$\hat{\mathbf{w}}_t = \nabla_{\mathbf{u}}(\gamma V_\pi(f(\mathbf{x}_t, \mathbf{u}) + \mathbf{w}_t) - (Q_\pi(\mathbf{x}_t, \mathbf{u}) - c(\mathbf{x}_t, \mathbf{u}))|_{\mathbf{u}=\mathbf{u}_t} ,$$

where $f(\mathbf{x}_t, \mathbf{u}) + \mathbf{w}_t$ represents the counterfactual next state after playing $\mathbf{u}$ at state $\mathbf{x}_t$. Note, this signal is a gradient of the temporal-difference error [Sutton and Barto, 2018], in fact being syntactically similar to expected SARSA. If $\mathbf{w}_t$ was in fact (*iid*) stochastic with $V_\pi, Q_\pi$ as corresponding value functions, this term on expectation would be zero. Therefore, this signal on average measures deviation introduced in $\mathbf{x}_{t+1}$ due to arbitrary or adversarial $\mathbf{w}_t$. We can also view this expression as

$$\hat{\mathbf{w}}_t = \nabla_{\mathbf{u}}(\gamma V_\pi(f(\mathbf{x}_t, \mathbf{u}) + \mathbf{w}_t) - \gamma V_\pi(f(\mathbf{x}_t, \mathbf{u})))|_{\mathbf{u}=\mathbf{u}_t} .$$

$V_\pi$ is quadratic in the linear quadratic regulator setting, so this becomes a linear function of $\mathbf{w}_t$. Computing $\nabla_{\mathbf{u}} V_\pi(f(\mathbf{x}_t, \mathbf{u}) + \mathbf{w}_t)|_{\mathbf{u}=\mathbf{u}_t}$ analytically would require knowledge of the dynamics, but luckily this can be efficiently estimated online. Using a policy $\pi$, with noised actions $\mathbf{u}_t = \pi(\mathbf{x}_t) + \mathbf{n}_t$, for $\mathbf{n}_t \sim \mathcal{N}(0, \Sigma)$ we have the following PD estimates:

$$\hat{\mathbf{w}}_t = \gamma V_\pi(\mathbf{x}_{t+1})\Sigma^{-1}\mathbf{n}_t - \nabla_{\mathbf{u}}(Q_\pi(\mathbf{x}_t, \mathbf{u}) - c(\mathbf{x}_t, \mathbf{u}))|_{\mathbf{u}=\mathbf{u}_t} , \tag{1}$$

$$\hat{\mathbf{w}}_t = (c(\mathbf{x}_t, \mathbf{u}_t) + \gamma V_\pi(\mathbf{x}_{t+1}) - Q_\pi(\mathbf{x}_t, \mathbf{u}_t))\Sigma^{-1}\mathbf{n}_t . \tag{2}$$

These are zeroth-order gradient estimators (see [Liu et al., 2020] for a more detailed exposition). Intuitively, the second estimator may have lower variance as the expected SARSA error can be much smaller than the magnitude of the value function. An additional benefit is that this implementation only requires a scalar cost signal without needing access to a differentiable cost function.

The most important property of this estimator is that it, in expectation, it produces a signal that is a linearly transformation of the true disturbance if the underlying setting is a linear dynamical system. This is formalized in the following lemma.

**Lemma 1.** *Consider a time-invariant linear dynamical systems with system matrices $A, B$ and quadratic costs, along with a linear baseline policy $\pi$ defined by control law $\mathbf{u}_t = -K_\pi \mathbf{x}_t$. In expectation, the pseudo disturbances (1) and (2) are linear transformations of the actual perturbation*

$$\mathbb{E}[\hat{\mathbf{w}}_t | \mathbf{x}_t] = T\mathbf{w}_t,$$

*where $T$ is a fixed linear operator that depends on the system.*

### 3.2 Pseudo-Disturbance Class II: Vector Value Functions

The second approach derives a signal from auxiliary value functions. Concretely, instead of scalar-valued cost function $c : \mathbb{R}^{d_x} \to \mathbb{R}$, consider a vector-valued cost function $\mathbf{c} : \mathbb{R}^{d_x} \to \mathbb{R}^{d_w}$. For such vector-valued cost, we introduce vectorized value and state-action value functions as

$$V_\pi^{\mathbf{c}} : \mathbb{R}^{d_x} \to \mathbb{R}^{d_w} \;,\; Q_\pi^{\mathbf{c}} : \mathbb{R}^{d_x} \times \mathbb{R}^{d_u} \to \mathbb{R}^{d_w} \;.$$

In particular, we have

$$Q_\pi^{\mathbf{c}}(\mathbf{x}, \mathbf{u}) = \mathbb{E}\left[\sum_{t=0}^{\infty} \gamma^t \mathbf{c}(\mathbf{x}_t^\pi) \,\bigg|\, \mathbf{x}_0^\pi = \mathbf{x}, \mathbf{u}_0^\pi = \mathbf{u}\right] \;, V_\pi^{\mathbf{c}}(\mathbf{x}) = \mathbb{E}\left[\sum_{t=0}^{\infty} \gamma^t \mathbf{c}(\mathbf{x}_t^\pi) \,\bigg|\, \mathbf{x}_0^\pi = \mathbf{x}\right] \;.$$

Our PD signal is then

$$\hat{\mathbf{w}}_t = \mathbf{c}(\mathbf{x}_t) + \gamma \mathbf{V}_\pi^{\mathbf{c}}(\mathbf{x}_{t+1}) - \mathbf{Q}_\pi^{\mathbf{c}}(\mathbf{x}_t, \mathbf{u}_t) \;. \tag{3}$$

In contrast to the first approach, for a fixed set of cost functions, this approach provides a deterministic PD-signal. This is very beneficial, as at inference time the DAC policy can be run without injecting additional noise and without requiring a high variance stochastic signal. This does come at a cost, as this method requires simultaneous off-policy evaluation for many auxiliary value functions (each corresponding to a different scalar cost) before DAC can be run via $Q$-function evaluations at inference, both of which can be significantly more expensive than the first approach.

For the case of linear dynamical systems, if we use *linear* costs on top of a linear base policy, this approach can recover the disturbances up to a linear transformation. It can be seen that the values corresponding to a linear cost function $c$ are linear functions of the state, and hence the vectorized versions are also linear functions of state. We formalize this as follows:

**Lemma 2.** *Consider a time-invariant linear dynamical systems with system matrices $A, B$, along with a linear baseline policy $\pi$ defined by control law $\mathbf{u}_t = -K_\pi \mathbf{x}_t$. Let $\mathbf{V}_\pi^{\mathbf{c}}$ and $\mathbf{Q}_\pi^{\mathbf{c}}$ be value functions for $\pi$ for i.i.d. zero mean noise with linear costs $\mathbf{c}(x) := Lx$, then the PD-signal (3) is a linear transformation*

$$\hat{\mathbf{w}}_t = T\mathbf{w}_t,$$

*where $T$ is a fixed linear operator that depends on the system and baseline policy $\pi$. In addition, if $L$ is full rank and the closed loop dynamics are stable, then $T$ is full rank.*

### 3.3 Pseudo-Disturbance Class III: Simulator Based

The last Pseudo-Disturbance signal we consider requires a potentially inaccurate simulator. It is intuitive, particularly simple to implement, and yet comes with theoretical guarantees.

The Pseudo-Disturbance is taken to be the difference between the actual state reached in an environment, and the expected state, over the randomness in the environment. To compute the expected state, we require the simulator $f_{\text{sim}}$ initialized at the current state. Formally,

$$\hat{\mathbf{w}}_t = \mathbf{x}_{t+1} - f_{\text{sim}}(\mathbf{x}_t, \mathbf{u}_t). \tag{4}$$

The simplicity of this PD is accompanied by a simple lemma on its characterization of the disturbance in a dynamical system, even if that system is time varying, as follows,

**Lemma 3.** *Suppose we have a simulator $f_{sim}$ such that $\forall \mathbf{x}, \mathbf{u}, \|f_{sim}(\mathbf{x}, \mathbf{u}) - f(\mathbf{x}, \mathbf{u})\| \leq \delta$, then Pseudo-Disturbance (4) is approximately equal to the actual perturbation $\|\widehat{\mathbf{w}}_t - \mathbf{w}_t\| \leq \delta$.*

### 3.4 Merits of different Pseudo-Disturbance signals

Each of the three PD signals described in this section offers something a bit different. PD3 offers the most direct disturbance signal, but comes with the requirement of a simulator. If the simulator is very accurate, this is likely the strongest signal, though this method may not be suitable with a large sim-to-real gap. PD1 and PD2 on the other hand, do not require a simulator but also have a natural trade off. PD1 is simpler and easier to add on top of an existing policy. However, it uses zeroth-order estimation, so the guarantees only hold in expectation and it may have high variance. On the other hand, PD2 is not a stochastic estimate, but it requires auxiliary value estimation from the base policy. This may come at the cost of additional space and computational complexity. In many cases, this can be handled using the same deep Q-network except with a wider head, which may not be so onerous. We note that PD2 **does not** require specific domain engineered signals for the auxiliary rewards. For example, using the coordinates of the state representation was enough to demonstrate improvements over baselines in our experiments. For richer, higher dimensional (visual) state spaces, this can be generalized using neural representations of state as the auxiliary reward, achieved by modulating the PD2 disturbance dimension to account for the fact that the underlying dynamics are simpler.

## 4 Meta Algorithm and Main Theorem

In this section we define a meta-algorithm for general reinforcement learning. The algorithm takes as an input an existing RL method, that may or may not have theoretical guarantees. It adds an additional layer on top, which estimates the Pseudo-Disturbances according to one of the three methods in the previous section. It then uses an online gradient method to optimize a linear policy in the past Pseudo-Disturbances. This can be viewed as a zeroth-order model-free version of the Gradient Perturbation Controller (GPC) [Agarwal et al., 2019].

The algorithm is formally defined in Algorithm 1. A typical choice of the parametrization $\pi(\cdot|M)$ is a linear function of a window of past disturbances (ie. Disturbance Action Control [Agarwal et al., 2019]).

$$\pi(\mathbf{w}_{t-1:t-h}|M_{1:h}) = \sum_{i=1}^{h} M_i \mathbf{w}_{t-i}. \tag{5}$$

---

**Algorithm 1** MF-GPC (Model-Free Gradient Perturbation Controller)

---

1: Input: Memory parameter $h$, learning rate $\eta$, exploration noise covariance $\Sigma$, initialization $M_{1:h}^1 \in \mathbb{R}^{d_u \times d_w \times h}$, initial value and $Q$ functions, base RL algorithm $\mathcal{A}$.

2: **for** $t = 1 \dots T$ **do**

3:    Use action $\mathbf{u}_t = \pi_{\text{base}}(\mathbf{x}_t) + \pi(\hat{\mathbf{w}}_{t-1:t-h}|M^t) + \mathbf{n}_t$, where $\mathbf{n}_t$ is *iid* Gaussian, i.e.

$$\mathbf{n}_t \sim \mathcal{N}(0, \Sigma)$$

4:    Observe state $\mathbf{x}_{t+1}$, and cost $c_t = c_t(\mathbf{x}_t, \mathbf{u}_t)$.

5:    Compute Pseudo-Disturbance [see (2),(3), (4)]

$$\hat{\mathbf{w}}_t = \text{PD-estimate}(\mathbf{x}_{t+1}, \mathbf{x}_t, \mathbf{u}_t, c_t, \mathbf{n}_t).$$

6:    Update policy parameters using the stochastic gradient estimate (see Section 4.1)

$$M^{t+1} \leftarrow M^t - \eta \, c_t(\mathbf{x}_t, \mathbf{u}_t) \Sigma^{-1} \sum_{j=0}^{h-1} \mathbf{n}_{t-i} \otimes J_i^t,$$

where $\otimes$ is an outer product and $J_i^t = \hat{\mathbf{w}}_{t-i-1:t-h-i}$ for (5), and more generally,

$$J_i^t = \left. \frac{\partial \pi(\hat{\mathbf{w}}_{t-i-1:t-h-i}|M_i)}{\partial M} \right|_{M=M^t}.$$

7: **end for**

8: Optionally, update the policy $\pi_{\text{base}}$ and its $Q, V$ functions using $\mathcal{A}$ so that they are Bellman consistent, i.e. they satisfy the policy version of Bellman equation.

---

We prove the following theorem for the case of linear dynamics:

**Theorem 4** (Informal Statement (see Theorem 8)). *If the underlying dynamics are linear with the state evolution specified as*

$$\mathbf{x}_{t+1} = A\mathbf{x}_t + B\mathbf{u}_t + \mathbf{w}_t,$$

*with $d_{\min} = \min\{d_x, d_u\}$, then then as long as the Pseudo-Disturbance signal $\hat{\mathbf{w}}_t$ satisfies $\hat{\mathbf{w}}_t = T\mathbf{w}_t$, for some (possibly unknown) invertible map $T$, Algorithm 1 generates controls $\mathbf{u}_t$ such that for any sequence of bounded (even adversarial) $\mathbf{w}_t$ such that the following holds*

$$\sum_t c_t(\mathbf{x}_t, \mathbf{u}_t) \;\leq\; \min_{\pi \in \Pi^{DAC}} \sum_t c_t(\mathbf{x}_t^\pi, \mathbf{u}_t^\pi) + \widetilde{\mathcal{O}}(\sqrt{d_u d_{\min}} T^{3/4}),$$

*for any any sequence of convex costs $c_t$, where the policy class $DAC$ refers to all policies $\pi$ that produce a control as a linear function of $\mathbf{w}_t$. Further, if the costs $c_t$ are L-smooth, the regret for Algorithm 1 admits an improved upper bound of $\widetilde{\mathcal{O}}((d_u d_{\min} T)^{2/3})$.*

In particular, the above theorem implies the stated regret bounds when the Pseudo-Disturbance is estimated as described in Equations 3 (Vector Value Function-based) and 4 (Simulator-based).

The regret bounds in Theorem 4 are strict improvements over state-of-the-art bounds in terms of dimension dependence; the latter operate with explicit descriptions of disturbances. This is achieved by using a better choice of gradient estimator, using exploration in action-space rather than parameter-space. As a result, our bounds have no dependence on the state dimension since $d_{\min} \leq d_u$. As an instructive case, for high-dimensional underactuated systems, where $d_u < d_x$, our regret bounds scale as $\tilde{O}(d_u T^{3/4})$ in contrast to $\tilde{O}(d_u d_x^{1/2} T^{3/4})$ for convex costs from [Gradu et al., 2020, Cassel and Koren, 2020], and as $\tilde{O}(d_u^{4/3} T^{2/3})$ for smooth costs improving over $\tilde{O}(d_u^{4/3} d_x^{2/3} T^{2/3})$ from [Cassel and Koren, 2020]. Note that the ratio by which we improve here can be unbounded, with larger improvements for high-dimensional ($d_x \gg 1$) systems. See Appendix E.2 for further details, comparisons and proofs.

### 4.1 Derivation of update

In the algorithm, the key component is computing an approximate policy gradient of the cost. A complete theoretical analysis of our algorithm can be found in Appendix E, but we provide a brief sketch of the gradient calculation. Let $J_t(M)$ denote the expected counterfactual cost $c_t$ of following policy $M$ with the same observed disturbances $w_t$. We first note that if the dynamics are suitably stabilized (which should be done by $\pi_{\text{base}}$), the state and cost can be approximated as a function $C$ of a small window of previous controls.

$$J_t(M) = \mathbb{E}_{\mathbf{n}_{1:t}}[c_t(\mathbf{x}_t^M, \mathbf{u}_t^M)] \approx \mathbb{E}_{\mathbf{n}_{t-h:t}}[C(\mathbf{u}_t(M) + \mathbf{n}_t, \ldots, \mathbf{u}_{t-h}(M) + \mathbf{n}_{t-h})],$$

where we use $u_{t-i}(M)$ as a shorthand for $\pi(\hat{\mathbf{w}}_{t-i-1:t-h-i}|M)$. The expression here is that of a Gaussian smoothed function, which allows us to get the following unbiased single point gradient estimate

$$\nabla_{\mathbf{u}_i} \mathbb{E}_{\mathbf{n}_{t-h:t}}[C(\mathbf{u}_t + \mathbf{n}_t, \ldots, \mathbf{u}_{t-h} + \mathbf{n}_{t-h})] = \mathbb{E}_{\mathbf{n}_{t-h:t}}[\Sigma^{-1} C(\mathbf{u}_t + \mathbf{n}_t, \ldots, \mathbf{u}_{t-h} + \mathbf{n}_{t-h})\mathbf{n}_i].$$

We use a single sample to get a stochastic gradient. Using the chain rule, which involves an outer product due to the tensor structure of $M$, we get stochastic gradients with respect to $M$ as follows

$$\widehat{\nabla_M J_t}(M) \approx C(\mathbf{u}_t(M) + \mathbf{n}_t, \ldots, \mathbf{u}_{t-h}(M) + \mathbf{n}_{t-h})\Sigma^{-1} \sum_{i=0}^{h-1} \mathbf{n}_{t-i} \otimes \frac{\partial \pi(\hat{\mathbf{w}}_{t-i-1:t-h-i}|M)}{\partial M}.$$

Finally, we note that $M^t$ is slowly moving because of gradient descent, so we can approximate

$$c_t(\mathbf{x}_t, \mathbf{u}_t) \approx C(\mathbf{u}_t(M^t) + \mathbf{n}_t, \ldots, \mathbf{u}_{t-h}(M^t) + \mathbf{n}_{t-h}).$$

Putting everything together, we have

$$\widehat{\nabla_M J_t}(M)\Big|_{M=M^t} \approx c_t(\mathbf{x}_t, \mathbf{u}_t)\Sigma^{-1} \sum_{i=0}^{h-1} \mathbf{n}_{t-i} \otimes \frac{\partial \pi(\hat{\mathbf{w}}_{t-i-1:t-h-i}|M)}{\partial M}\Big|_{M=M^t}. \tag{6}$$

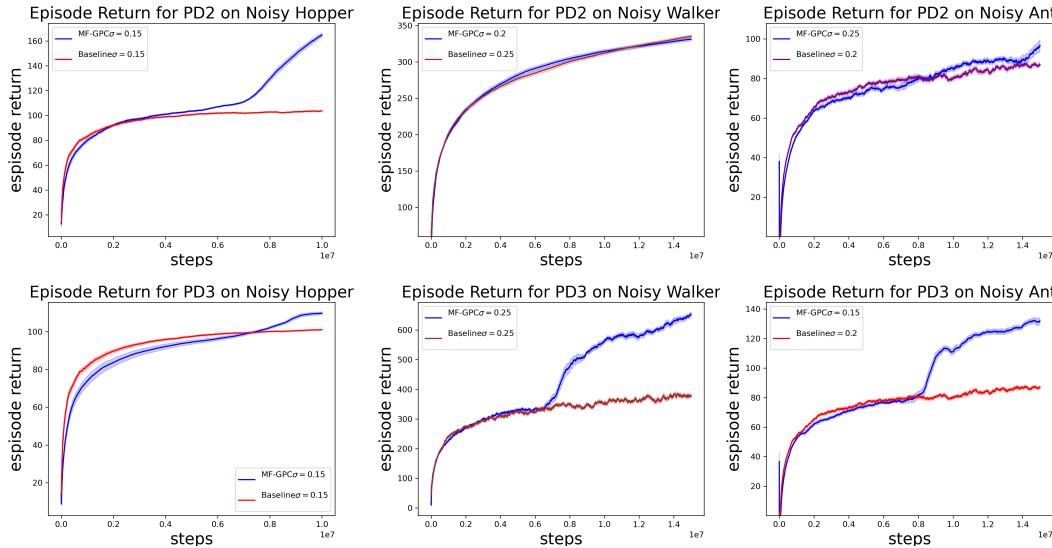

Figure 1: Episode return for best performing MF-GPC model versus best performing baseline DDPG model for various OpenAI Gym environments and pseudo-estimation methods. Environment and pseudo-estimation method shown in title. Results averaged over 25 seeds. Shaded areas represent confidence intervals. We find that PD2 and PD3 perform well in these settings.

## 5    Experiments

We apply the MF-GPC Algorithm 1 to various OpenAI Gym [Brockman et al., 2016] environments. We conduct our experiments in the research-first modular framework Acme [Hoffman et al., 2020]. We pick $h = 5$ and use the DDPG algorithm [Lillicrap et al., 2016] as our underlying baseline. We update the $M$ matrices every 3 episodes instead of continuously to reduce runtime. We also apply weight decay to line 6 of Algorithm 1. Our implementation of PD1 is based on Equation 2. PD2 can be implemented with any vector of rewards. We choose linear function $L$ given in Lemma 2 to be the identity function. Hence $\mathbf{c}$ in Equation 3 reduces to the state $x_t$ itself. We pick $\mathbf{V}$ and $\mathbf{Q}$ network architectures to be the first $d_x$ units of the last layer of the critic network architecture. We train for 1e7 steps as a default (this is also the default in the Acme code) and if performance has not converged we extend to 1.5e7 steps. Because the $M$ matrices impact the exploration of the algorithm, we tune the exploration parameter $\sigma$ for both DDPG and MF-GPC. For the baseline DDPG, we typically explore $\sigma \in \{0.15, 0.2, 0.25\}$. More experimental details may be found in Appendix Section B.

**Results for Noisy Hopper, Walker 2D, and Ant**    We create a noisy Hopper, Walker 2D, and Ant environments by adding a Uniform random variable $U[-0.1, 0.1]$ to the state. The noise is added at every step for both the DDPG baseline and our MF-GPC. We plot the results for PD2, and PD3 in Figure 1. We find that PD2 and PD3 perform relatively well in these settings. Graphs depicting all runs for different $\sigma$ are available in Appendix Section B. MF-GPC is not guaranteed to improve performance in realistic RL settings. We find that generally PD1 does not perform well e.g. in Figure 2 a) and some examples where applying it yields performance similar to baseline are given in Appendix Section B. This is likely due to the high variance of the PD estimate. We find that neither our method nor the baseline is too sensitive to our hyper-parameter tuning (Figure 2 b) ), possibly because we start with the default Acme parameters which are already well tuned for the noiseless environment.

**Linear Dynamical Systems**    We evaluate our methods on both low dimensional ($d_x = 2, d_u = 1$) and a higher dimensional ($d_x = 10, d_u = 5$) linear systems with sinusoidal disturbances to demonstrate the improvements in dimension of our method (labeled RBPC) over BPC [Gradu et al., 2020]. We use the full information GPC [Agarwal et al., 2019] and LQR as baselines using implementations from Gradu et al. [2021]. While performance is comparable to BPC on the small system, on the larger system, BPC could not be tuned to learn while RBPC improves upon the LQR

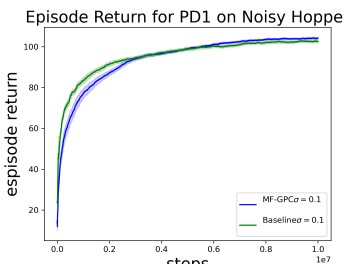 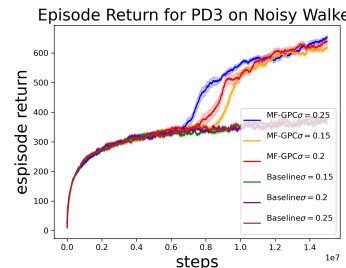

Figure 2: Left: Episode return for PD1 for Noisy Hopper. We find that PD1 is not effective for RL settings. Right: Hyper-parameter search for PD3 on Noisy Walker. We find that neither Meta-GPC nor the baseline DDPG algorithm is too sensitive to tuning.

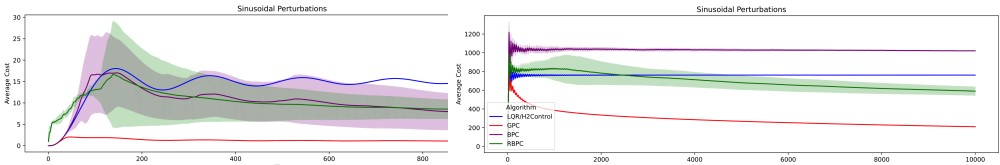

Figure 3: Comparison on low dimensional (left) vs high dimensional (rights) LDS.

baseline (see Figure 3). In both experiments, $h = 5$ and the learning rate and exploration noise is tuned.

## 6    Conclusion

We have described a new approach for model-free RL based on recent exciting advancements in model based online control. Instead of using state-based policies, online nonstochastic control proposes the use of disturbance-based policies. To create a disturbance signal without a model, we define three possible signals, called Pseudo-Disturbances, each with its own merits and limitations. We give a generic (adaptable) REINFORCE-based method using the PD signals with provable guarantees: if the underlying MDP is a linear dynamical system, we recover and improve the strong guarantees of online nonstochastic control. Preliminary promising experimental results are discussed. We believe this is a first step in the exciting direction of applying tried-and-tested model-based control techniques for general reinforcement learning.

## Acknowledgments and Disclosure of Funding

Elad Hazan acknowledges funding from the ONR award N000142312156, the NSF award 2134040, and Open Philanthropy.

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

# Contents

# A Notation

We use the following notation consistently throughout the paper:

| Symbol | Semantics |
|---|---|
| $\otimes$ | outer product |
| $f$ | dynamics/transition function |
| $f_{\text{sim}}$ | simulator dynamics/transition function |
| $d_x$ | state dimension |
| $d_u$ | control dimension |
| $d_w$ | pseudo-disturbance |
| $d_{\min}$ | $\min(d_x, d_u)$ |
| $\mathbf{x}_t \in \mathbb{R}^{d_x}$ | state at time $t$ |
| $\mathbf{u}_t \in \mathbb{R}^{d_u}$ | control at time $t$ |
| $\mathbf{w}_t \in \mathbb{R}^{d_w}$ | perturbation (disturbance) at time $t$ |
| $c_t$ | instantaneous cost at time $t$ |
| $\widehat{\mathbf{w}}_t \in \mathbb{R}^{d_w}$ | pseudo-disturbance at time $t$ |
| $\mathbf{n}_t \in \mathbb{R}^{d_u}$ | Gaussian exploration noise at time $t$ |
| $A, B, C$ | system matrices for linear dynamical system |
| $h$ | history length (i.e., number of parameters) in a policy class |
| $M_{1:h}^t$ | $h$-length sequence of matrices used by MF-GPC at time $t$ |
| $\mathcal{M}$ | policy class of $h$-length matrices |
| $\gamma$ | discount factor |
| $V_\pi, Q_\pi$ | state and state-action value functions for $\pi$ |
| $\mathbf{V}_\pi^{\mathbf{r}}, \mathbf{Q}_\pi^{\mathbf{r}}$ | vectorized value and $Q$ functions for $\pi$ for reward vectors $\mathbf{r}(x)$ |
| $y_t$ | idealized state |
| $\tilde{C}_t$ | stationary idealized cost (function of single $M$) at time $t$ |
| $C_t$ | non-stationary idealized cost (function of memory) at time $t$ |
| $C_{t,\delta}$ | Smoothed $C_t$ using noised controls |
| $F_t$ | idealized cost as a function of last controls at time $t$ |
| $F_{t,\delta}$ | smoothed $F_t$ |
| $\|\cdot\|$ | spectral norm |
| $\|\cdot\|_F$ | Frobenius norm |

# B Experiments

We test the performance of our method on various OpenAI Gym environments. We conduct our experiments in the research-first modular framework Acme [Hoffman et al., 2020]. We pick $h = 5$ and use the DDPG algorithm [Lillicrap et al., 2016] as our underlying baseline. We update the $M$ matrices every 3 episodes instead of continuously to reduce runtime. We also apply weight decay to line 6 of Algorithm 1.

Our implementation is based on the Acme implementation of D4PG. The policy and critic networks both have the default sizes of $256 \times 256 \times 256$. We use the Acme default number of atoms as 51 for the network. We run in the distributed setting with 4 agents. The underlying learning rate of the D4PG implementation is left at $3e - 04$. The exploration parameter, $\sigma$ is tuned.

**Plotting** We use a domain invariant exponential smoother with a small smoothing parameter of 0.1 . The smoothing is applied before the mean is taken over the data. To construct the confidence intervals, we take the following steps 1) smooth the data 2) linearly interpolate each run of the data to produce a fine grid of values 3) calculate $\sigma/\sqrt{N}$ on interpolated data.

## B.1 Hopper

The OpenAI Gym Hopper environment is a two dimensional one legged figure that consists of four body parts, namely a torso, thigh, leg and foot.

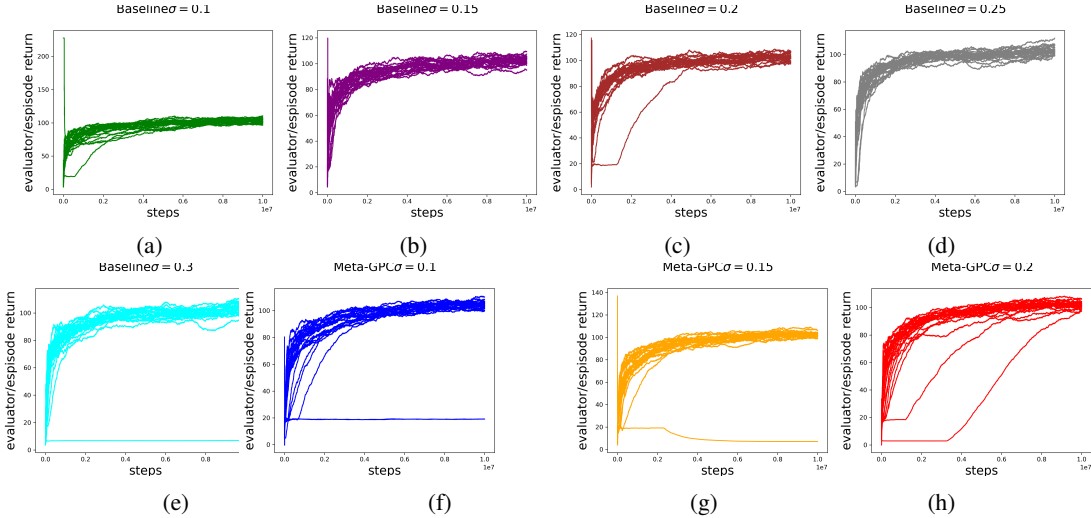

Figure 4: Raw data for PD Estimate 1 on Hopper. Each plot represents either the baseline or our method for a different setting of the exploration parameter, $\sigma$. We find that there are some outlier runs (for example the horizontal line in subfig f). We remove the two lowest return runs from each group before plotting.

## B.2 Noisy Hopper

We create a Noisy Hopper environment by adding a Uniform random variable $U[-0.1, 0.1]$ to qpos during training and evaluation. The noise is added at every step, and both DDPG and MF-GPC are evaluated on the same noisy environment.

**PD1** We implement PD1 for Noisy Hopper according to Equation 2. We see our results in Figure 6 in the first column. We tune $\sigma$ for the D4PG baseline in the set $\{0.1, 0.15, 0.2, 0.25, 0.3\}$. We find that the $\sigma = 0.1$ performs the best. We tune $\sigma$ for MF-GPC in the set $\{0.1, 0.15, 0.2\}$. We find that the $\sigma = 0.1$ setting performs the best. We averaged our results over 25 seeds. Solid line represents the mean return over all runs and shaded areas represent standard error. We find that MF-GPC run with PD-1 has a small advantage compared to the DDPG baseline on Noisy Hopper. **Removing outliers for PD1** for this specific experiment, we notice that some runs seem to be outliers. Therefore, when plotting, we remove the lowest 2 runs across all groups (of both the baseline and our method). Complete raw data (with outliers included) can be seen in Figure 4.

**PD-2** PD2 can be implemented with any vector of rewards. We choose linear function $L$ given in Lemma 2 to be the identity function. Hence $\mathbf{c}$ in Equation 3 reduces to the state $x_t$ itself. We pick $\mathbf{V}$ and $\mathbf{Q}$ to be the first $d_x$ units of the last layer of the critic network. If $d_x$ is larger than the number of atoms of the critic network (51) we take all 51 nodes from the critic network. We find that a default $\sigma = 0.15$ performs well for MF-GPC so we do not tune $\sigma$ further.

**PD-3** In practice the expectation in Equation 4 requires estimation. We use an average over 4 copies of the environment for this estimation.

## B.3 Noisy Walker 2D and Ant

We follow the basic procedure for Hopper but train for 15 million steps instead of 10 million. We report our results for the hyper-parameter sweeps in columns 2 and 3 of Figure 5. We find that PD2 and PD3 perform relatively well in these settings.

**PD-2** PD2 can be implemented with any vector of rewards. We choose linear function $L$ given in Lemma 2 to be the identity function. Hence $\mathbf{c}$ in Equation 3 reduces to the state $x_t$ itself. Recall that $d_x$ is the dimension of the state space. We pick $\mathbf{V}$ and $\mathbf{Q}$ to be the first $d_x$ units of the last layer of the critic network. If $d_x$ is larger than the number of atoms of the critic network (51) we take all 51 nodes from the critic network.

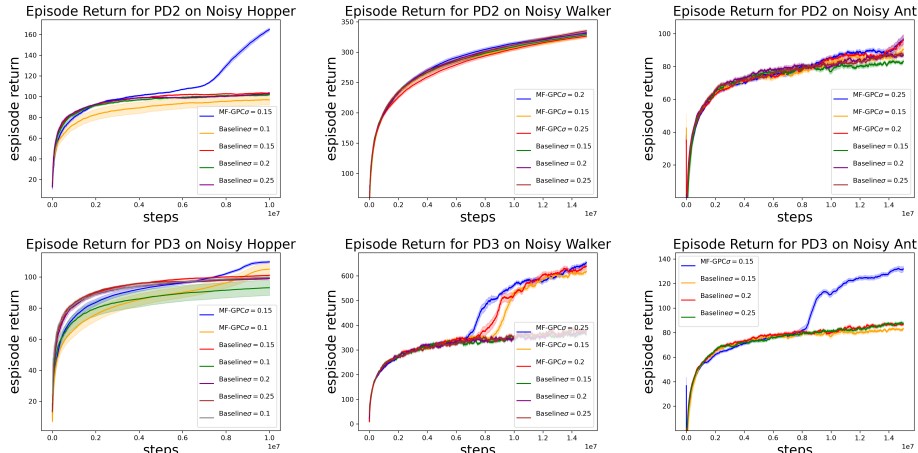

Figure 5: Episode return for best performing MF-GPC model versus best performing baseline DDPG model for various OpenAI Gym environments and pseudo-estimation methods. Environment and pseudo-estimation method shown in title. Results averaged over 25 seeds. Shaded areas represent confidence intervals. We find that PD2 and PD3 perform well in these settings.

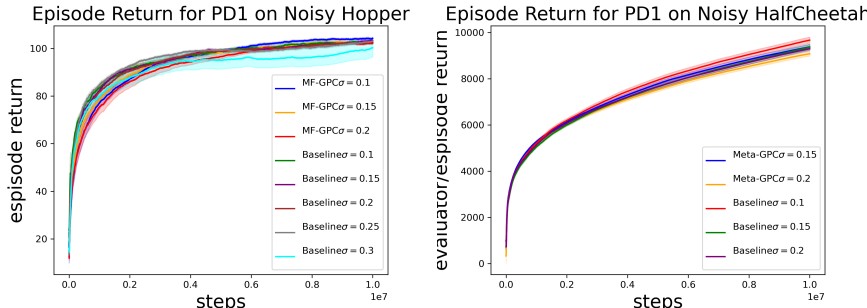

Figure 6: Left: Episode return for PD1 for Noisy Hopper. Right: Episode return for PD1 for Noisy Half Cheetah. We find that PD1 is not effective for RL settings.

**PD-3** In practice the expectation in Equation 4 requires estimation. We use an average over 4 copies of the environment for this estimation. For Noisy Ant, we find that a default $\sigma = 0.15$ performs well for MF-GPC so we do not tune $\sigma$ further.

## B.4 Experiments with adversarial noise

We run MF-GPC on top of a DDPG baseline for the inverted pendulum environment with 1) fast gradient sign method noise [Goodfellow et al., 2014] and 2) noise from a discretized sinusoid. We plot our results in Figure 7

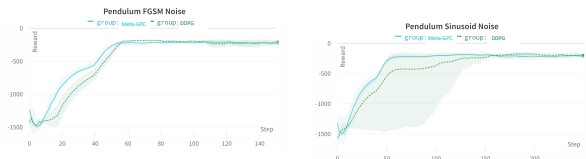

Figure 7: Results from our method on the inverted pendulum environment for fast gradient sign method noise and sinusoid noise.

# C  Discrete State and Action Spaces

In this section, we consider differentiable parameterized random policies. For finite action space $\mathcal{U}$, let $\Delta^{\mathcal{U}} = \{p : \mathcal{U} \to [0,1] | \sum_{a \in \mathcal{U}} p(a) = 1\}$ be the probability simplex over $\mathcal{U}$. Our policies, will be parameterized by $M$ and depend on a window of past Pseudo-disturbances, providing the following distribution over actions.

$$\mathbf{u} \sim \pi(\cdot | \hat{\mathbf{w}}_{t-1:t-h}, M) \in \Delta^{\mathcal{U}} \tag{7}$$

The baseline policy, would be built into $\pi$ in this setting. For example, we could have a softmax neural net policy, and our algorithm adds a residual correction to the logits.

**Implementation of PD signals in discrete spaces**  For discrete spaces, PD-2 defined via (3) is well defined, as we can still create auxiliary $Q$ functions in the discrete space to produce our signal. Because, we no longer use Gaussian noise, PD-1 (2) can be modified as follows:

$$\boxed{\hat{\mathbf{w}}_t = (c(\mathbf{x}_t, \mathbf{u}_t) + \gamma V_\pi(\mathbf{x}_{t+1}) - Q_\pi(\mathbf{x}_t, \mathbf{u}_t)) \nabla_M \log(\pi(\mathbf{u}_t | \hat{\mathbf{w}}_{t-h:t-1}, M))|_{M=M_t}} \tag{8}$$

For our zeroth order gradient, we use the REINFORCE:

---

**Algorithm 2** DMF-GPC (Discrete Model-Free Gradient Perturbation Controller)

---
1: Input: Memory parameter $h$, learning rate $\eta$, initialization $M_{1:h}^1$, initial value and $Q$ functions, base RL algorithm $\mathcal{A}$.
2: **for** $t = 1 \ldots T$ **do**
3:
$$\text{Sample action } \mathbf{u}_t \sim \pi(\cdot | \hat{\mathbf{w}}_{t-1:t-h}, M^t)$$
4:    Observe state $\mathbf{x}_{t+1}$, and cost $c_t = c_t(\mathbf{x}_t, \mathbf{u}_t)$.
5:    Compute Pseudo-Disturbance [see (3), (8)]
$$\hat{\mathbf{w}}_t = \text{PD-estimate}(\mathbf{x}_{t+1}, \mathbf{x}_t, \mathbf{u}_t, c_t).$$
6:    Update policy parameters using the stochastic gradient estimate
$$M^{t+1} \leftarrow M^t - \eta \, c_t(\mathbf{x}_t, \mathbf{u}_t) \sum_{j=0}^{h-1} \nabla_M \log(\pi(\mathbf{u}_{t-j} | \hat{\mathbf{w}}_{t-j-1:t-j-h}, M))|_{M=M_t} \ .$$
7: **end for**
8: Optionally, update the baseline policy parameters and its $Q, V$ functions using $\mathcal{A}$ so that they are Bellman consistent, i.e. they satisfy the policy version of Bellman equation.

---

# D  Pseudo-Disturbance Proofs

In this appendix, we have the deferred proofs from Section 3. For convenience, the lemmas have also been restated.

## D.1  Proof of Lemma 1

**Lemma 5.** *For time-invariant linear dynamical systems with system matrices $A$, $B$ and quadratic costs, in expectation, the pseudo disturbances (1) and (2) is a linear transformation of the actual perturbation*
$$\mathbb{E}[\hat{\mathbf{w}}_t | \mathbf{x}_t] = T\mathbf{w}_t,$$
*where $T$ is a fixed linear operator that depends on the system.*

*Proof.* Recall from the theory of the linear quadratic regulator that the value function of an infinite horizon LDS is quadratic Bertsekas [2012],

$$V(\mathbf{x}) = \mathbf{x}^\top P \mathbf{x}.$$

Thus,

$$\mathbb{E}[\nabla_{\mathbf{u}}(Q(\mathbf{x}_t, \mathbf{u}) - c(\mathbf{x}_t, \mathbf{u}))|_{\mathbf{u}=\mathbf{u}_t}] = \mathbb{E}[\gamma B^\top P(A\mathbf{x}_t + B\mathbf{u}_t)]$$
$$= \mathbb{E}[\gamma B^\top P(A\mathbf{x}_t + B\pi(\mathbf{x}_t) + B\mathbf{n}_t)]$$
$$= \gamma B^\top P(A\mathbf{x}_t + B\pi(\mathbf{x}_t)).$$

By the definition of the signal, we have that

$$\mathbb{E}[\hat{\mathbf{w}}_t|\mathbf{x}_t] = \gamma\mathbb{E}[V(\mathbf{x}_{t+1})\Sigma^{-1}\mathbf{n}_t - \nabla_{\mathbf{u}}(Q(\mathbf{x}_t, \mathbf{u}) - c(\mathbf{x}_t, \mathbf{u}))|_{\mathbf{u}=\mathbf{u}_t}]$$
$$= \gamma\mathbb{E}[V(\mathbf{x}_{t+1})\Sigma^{-1}\mathbf{n}_t|\mathbf{x}_t] - \gamma B^\top P(A\mathbf{x}_t + B\pi(\mathbf{x}_t)).$$

Writing the quadratic value function also to the first term, and denoting $\|\mathbf{x}\|_P^2 = \mathbf{x}^\top P\mathbf{x}$, we have that

$$\mathbb{E}[V(\mathbf{x}_{t+1})\Sigma^{-1}\mathbf{n}_t|\mathbf{x}_t] = \mathbb{E}[\|A\mathbf{x}_t + B\pi(\mathbf{x}_t) + \mathbf{w}_t + B\mathbf{n}_t\|_P^2\Sigma^{-1}\mathbf{n}_t|\mathbf{x}_t]$$
$$= \Sigma^{-1}\mathbb{E}[\mathbf{n}_t\mathbf{n}_t^\top]B^\top P(A\mathbf{x}_t + B\pi(\mathbf{x}_t) + \mathbf{w}_t)$$
$$= B^\top P(A\mathbf{x}_t + B\pi(\mathbf{x}_t) + \mathbf{w}_t)$$

We thus conclude,

$$\mathbb{E}[\hat{\mathbf{w}}_t|\mathbf{x}_t] = \gamma\mathbb{E}[V(\mathbf{x}_{t+1})\Sigma^{-1}\mathbf{n}_t|\mathbf{x}_t] - \gamma B^\top P(A\mathbf{x}_t + B\pi(\mathbf{x}_t))$$
$$= \gamma B^\top P(A\mathbf{x}_t + B\pi(\mathbf{x}_t) + \mathbf{w}_t) - \gamma B^\top P(A\mathbf{x}_t + B\pi(\mathbf{x}_t))$$
$$= \gamma B^\top P\mathbf{w}_t = T\mathbf{w}_t$$

as needed. $\square$

## D.2 Proof of Lemma 2

**Lemma 6.** *Consider a time-invariant linear dynamical systems with system matrices $A$, $B$, along with a linear baseline policy $\pi$ defined by control law $\mathbf{u}_t = -K_\pi\mathbf{x}_t$. Let $\mathbf{V}_\pi^{\mathbf{c}}$ and $\mathbf{Q}_\pi^{\mathbf{c}}$ be value functions for $\pi$ for i.i.d. zero mean noise with linear costs $\mathbf{c}(x) := Lx$, then the PD-signal (3) is a linear transformation*

$$\hat{\mathbf{w}}_t = T\mathbf{w}_t,$$

*where $T$ is a fixed linear operator that depends on the system and baseline policy $\pi$. In addition, if $L$ is full rank and the closed loop dynamics are stable, then $T$ is full rank.*

*Proof.* We first note that for linear rewards, value functions for i.i.d. zero mean noise are equivalent the value functions without noise. As such, we have the identity

$$\mathbf{Q}_\pi^{\mathbf{c}}(\mathbf{x}_t, \mathbf{u}_t) = \mathbf{c}(\mathbf{x}_t) + \gamma\mathbf{V}_\pi^{\mathbf{c}}(A\mathbf{x}_t + B\mathbf{u}_t),$$

and so, we can rewrite or PD-signal as

$$\hat{\mathbf{w}}_t = \mathbf{V}_\pi^{\mathbf{c}}(\mathbf{x}_{t+1}) - \mathbf{V}_\pi^{\mathbf{c}}(A\mathbf{x}_t + B\mathbf{u}_t)$$
$$= \mathbf{V}_\pi^{\mathbf{c}}(A\mathbf{x}_t + B\mathbf{u}_t + \mathbf{w}_t) - \mathbf{V}_\pi^{\mathbf{c}}(A\mathbf{x}_t + B\mathbf{u}_t).$$

Now, it remains to show that $\mathbf{V}_\pi^{\mathbf{c}}$ is a fixed linear transformation. Indeed, we have

$$\mathbf{V}_\pi^{\mathbf{c}}(x) = \sum_{t=0}^{\infty}\gamma^i\mathbf{c}(A_\pi^t x) = \sum_{t=0}^{\infty}\gamma^i LA_\pi^t x = L(I - \gamma A_\pi)^{-1}x,$$

where $A_\pi = A - BK_\pi$ is the closed loop dynamics matrix. We now have

$$\hat{\mathbf{w}}_t = \mathbf{V}_\pi^{\mathbf{c}}(A\mathbf{x}_t + B\mathbf{u}_t + \mathbf{w}_t) - \mathbf{V}_\pi^{\mathbf{c}}(A\mathbf{x}_t + B\mathbf{u}_t)$$
$$= L(I - \gamma A_\pi)^{-1}[(A\mathbf{x}_t + B\mathbf{u}_t + \mathbf{w}_t) - (A\mathbf{x}_t + B\mathbf{u}_t)]$$
$$= L(I - \gamma A_\pi)^{-1}\mathbf{w}_t.$$

Now, stability of $\pi$ dictates $(I - \gamma A_\pi)$ is full rank (even for $\gamma = 1$), so if $L$ is full rank, $L(I - \gamma A_\pi)^{-1}$ is full rank.

$\square$

### D.3  Proof of Lemma 3

**Lemma 7.** *Suppose we have a simulator $f_{sim}$ such that $\forall \mathbf{x}, \mathbf{u}, \|f_{sim}(\mathbf{x}, \mathbf{u}) - f(\mathbf{x}, \mathbf{u})\| \leq \delta$, then Pseudo-Disturbance (4) is approximately equal to the actual perturbation $\|\widehat{\mathbf{w}_t} - \mathbf{w}_t\| \leq \delta$.*

*Proof.* This Lemma is immediate from the definition of the dynamics as $\mathbf{x}_{t+1} = f(\mathbf{x}_t, \mathbf{u}_t) + \mathbf{w}_t$. $\quad\square$

## E  Main Result and Dimension-efficient Bandit GPC

Below, we formally state and prove the main result. Subsequent sections attest to the fact that this regret bound is an unconditional improvement over the best known Gradu et al. [2020], Cassel and Koren [2020] in terms of its dependence on dimension and applicability to high-dimensional systems, even for the well-studied setting of linear control.

### E.1  Main Result

Using Theorem 9 which we state and prove in subsequent sections, we can prove the main result.

**Theorem 8.** *Consider a modification of Algorithm 3 implemented using $\hat{\mathbf{w}}_t$ in place of $\mathbf{w}_t$ and choose the step size as $\eta = \sqrt{\frac{d_{min}}{d_u}} T^{-3/4}$, and the exploration radius as $\delta = \sqrt{d_u d_{min}} T^{-1/4}$.*

*If the underlying dynamics are linear and satisfy the assumptions in Section E.2 then then as long as the Pseudo-Disturbance signal $\hat{\mathbf{w}}_t$ satisfies $\hat{\mathbf{w}}_t = \mathcal{T}\mathbf{w}_t$, for some (possibly unknown) invertible map $\mathcal{T}$, with $\mathbf{u}_t$ such that for any sequence of bounded (even adversarial) $\mathbf{w}_t$ such that the following holds*

$$\sum_t c_t(\mathbf{x}_t, \mathbf{u}_t) - \inf_{M \in \mathcal{M}} \sum_t c_t(\mathbf{x}_t^M, \mathbf{u}_t^M) \leq \widetilde{\mathcal{O}}(\sqrt{d_u d_{min}}\,\text{poly}(\|\mathcal{T}\|\|\mathcal{T}^{-1}\|)T^{3/4}),$$

*for any any sequence of convex costs $c_t$. Further, if the costs $c_t$ are L-smooth, the regret can be improved upper bound of*

$$\sum_t c_t(\mathbf{x}_t, \mathbf{u}_t) - \inf_{M \in \mathcal{M}} \sum_t c_t(\mathbf{x}_t^M, \mathbf{u}_t^M) \leq \widetilde{\mathcal{O}}(\text{poly}(\|\mathcal{T}\|\|\mathcal{T}^{-1}\|)(d_u d_{min} T)^{2/3}).$$

*Proof.* This follows from that fact that an invertible linear transformation $\mathcal{T}$ of $\mathbf{w}_t$ does not change the expressiveness of a DAC policy class $\mathcal{M}$ except through constants related to the norm of $\mathcal{T}$. More specifically, given a DAC policy $M_{1:h}$ that acts on true disturbances $\mathbf{w}_s$, the same exact controls are produced by a DAC policy $M'$ with $M'_i = M_i \mathcal{T}^{-1}$ acting on $\hat{\mathbf{w}}_s = \mathcal{T}\mathbf{w}_s$. The disturbances are also scaled by $\mathcal{T}$. by As such, we can attain equivalent regret bounds with a new policy class $\mathcal{M}'$ with diameter scaled by $\|\mathcal{T}^{-1}\|$ and new bound on disturbances $W' = \|\mathcal{T}\|W$. In Theorem 9, the hidden dependence on the DAC diameter and disturbance size are polynomial, yielding at most $\text{poly}(\|\mathcal{T}\|\|\mathcal{T}^{-1}\|)$ scaling in the regret. $\quad\square$

### E.2  Dimension-Efficient Bandit GPC

Under bandit feedback, the learner can only observe the cost it incurs, and does not have access to function value oracles or gradients of the cost functions. This setting has been studied in detail in linear control subject to adversarial disturbances using both dynamic and static regret settings; we restrict our attention to the latter.

A key characteristic of our proposed algorithm is that it performs exploration in the action space, rather than the policy space. This enables us to obtain a favorable trade-off between the quality of the proxy of the gradient and the amount of modification the objective (via randomized smoothing) is subject to. Leveraging this property, we show that our approach obtains a better regret bound than the best known [Cassel and Koren, 2020, Gradu et al., 2020] in the literature. In particular, the best known regret bounds for this setting scale as $O(\sqrt{d_x d_u d_{min}} T^{3/4})$. In contrast, we offer a regret bound of $O(\sqrt{d_u d_{min}} T^{3/4})$. This is both a quantitative and a qualitative improvement, and carries over to the case of smooth costs too. In particular, since our bound has no dependence on $d_x$ whatsoever, it is equally applicable to the high-dimensional setting ($d_x \gg 1$), which existing

methodologies fail to scale to in the bandit setting. We stress that this improvement in the upper bound stems from the *right* algorithm design, and not just a tighter analysis.

In this section, we analyze Algorithm 3, a minimally modified version of Algorithm 1, which uses delayed gradient updates. As a convention, we hold $\mathbf{w}_t = 0$ for $t < 0$ when defining DAC policies in early rounds.

---

**Algorithm 3** Bandit GPC

---

1: Input: Memory parameter $h$, learning rate $\eta$, exploration size $\delta$, initialization $M^1_{1:h} \in \mathbb{R}^{d_u \times d_x \times h}$, and convex policy set $\mathcal{M}$.
2: **for** $t = 1 \ldots T$ **do**
3:     Use action $\mathbf{u}_t = \sum_{i=1}^{h} M^t_i \mathbf{w}_{t-i} + \delta \mathbf{n}_t$, where $\mathbf{n}_t$ is drawn *iid* from a sphere uniformly , i.e.

$$\mathbf{n}_t \sim \text{Unif}(\mathbb{S}_{d_u}).$$

4:     Observe state $\mathbf{x}_{t+1}$, and cost $c_t = c_t(\mathbf{x}_t, \mathbf{u}_t)$.
5:     Store the stochastic gradient estimate

$$\widehat{\nabla}_t = \frac{d_u c_t(\mathbf{x}_t, \mathbf{u}_t)}{\delta} \sum_{i=0}^{h-1} \mathbf{n}_{t-i} \otimes \mathbf{w}_{t-i-1:t-i-h}$$

6:     Update using delayed gradient with euclidean projection onto $\mathcal{M}$

$$M^{t+1} = \Pi_{\mathcal{M}} \left[ M^t - \eta \widehat{\nabla}_{t-h} \right]$$

7: **end for**

---

We make the following assumptions pertaining to costs and linear dynamics:

1. The underlying dynamics are assumed to be time-invariant and linear, i.e.
$$\mathbf{x}_{t+1} = A\mathbf{x}_t + B\mathbf{u}_t + \mathbf{w}_t,$$
   where $\mathbf{x}_t, \mathbf{w}_t \in \mathbb{R}^{d_x}, \mathbf{u}_t \in \mathbb{R}^{d_u}, \|w_t\| \leq W$ and $\|B\| \leq \kappa$.
2. The linear system is $(\kappa, \alpha)$-strongly stable: $\exists Q, L$ such that
$$A = QLQ^{-1},$$
   where $\|Q^{-1}\|, \|Q\| \leq \kappa, \|L\| \leq 1 - \alpha$.
3. The time-varying online cost functions $c_t(\mathbf{x}_t, \mathbf{u}_t)$ are convex and satisfy for all $\|\mathbf{x}\|, \|\mathbf{u}\| \leq D$ that
$$c_t(\mathbf{x}, \mathbf{u}) \leq C \max\{D^2, 1\}, \quad \text{and} \quad \|\nabla_{\mathbf{u}} c(\mathbf{x}, \mathbf{u})\|, \|\nabla_{\mathbf{x}} c(\mathbf{x}, \mathbf{u})\| \leq G \max\{D, 1\}.$$
4. We define our comparator set $\mathcal{M} = \mathcal{M}_1 \times \cdots \times \mathcal{M}_h$ where
$$\mathcal{M}_i = \{M \in \mathbb{R}^{d_u \times d_x} : \|M\| \leq 2\kappa^4 (1-\alpha)^i\}$$
   as in [Cassel and Koren, 2020]. Let $d_{min} = \min\{d_x, d_u\}$.

The second assumption may be relaxed to that of stabilizability, the case when the linear system by itself might be unstable, however the learner is provided with a suboptimal linear controller $K_0$ such that $A + BK_0$ is $(\kappa, \alpha)$-strongly stable, via a blackbox reduction outlined in Proposition 6 (Appendix A) in Cassel et al. [2022].

**Theorem 9.** *Choosing the step size as $\eta = \sqrt{\frac{d_{min}}{d_u}} T^{-3/4}$, and the exploration radius as $\delta = \sqrt{d_u d_{min}} T^{-1/4}$, the regret of Algorithm 3 is upper bounded as*

$$Regret(\mathcal{A}) = \mathbb{E}\left[ \sum_{t=1}^{T} c_t(\mathbf{x}_t, \mathbf{u}_t) \right] - \inf_{M \in \mathcal{M}} \sum_{t=1}^{T} c_t(\mathbf{x}_t^M, \mathbf{u}_t^M) \leq \widetilde{\mathcal{O}}(\sqrt{d_u d_{min}} T^{3/4}).$$

*Furthermore, if the costs $c_t$ is L-smooth, then choosing $\delta = (d_u d_{min})^{1/3} T^{-1/6}, \eta = d_{min}^{1/3}/(d_u^{2/3} T^{2/3})$, the regret incurred by the algorithm admits an tighter upper bound of*

$$Regret(\mathcal{A}) = \mathbb{E}\left[ \sum_{t=1}^{T} c_t(\mathbf{x}_t, \mathbf{u}_t) \right] - \inf_{M \in \mathcal{M}} \sum_{t=1}^{T} c_t(\mathbf{x}_t^M, \mathbf{u}_t^M) \leq \widetilde{\mathcal{O}}((d_u d_{min} T)^{2/3}).$$

### E.3 Idealized Cost and Proof of Theorem 9

We will prove our result by creating a proxy loss with memory which accurately estimates the cost, showing that our update provides a low bias gradient estimate with suitably small variance. This will allow us to prove a regret bound on our proxy-losses, which we then translate to a regret bound on the policy itself.

Following [Agarwal et al., 2019, Cassel and Koren, 2020], we introduce a transfer matrix that describes the effect of recent disturbances on the state.

**Definition 10.** For any $i < 2h$, define the disturbance-state transfer matrix Let

$$\Psi_i(M^{1:h}) = A^i \mathbf{1}_{i \leq h} + \sum_{j=1}^{h} A^j B M_{i-j-1}^{h-j+1} \mathbf{1}_{i-j-1 \in [1,h]}$$

We can also create a transfer matrix for the effect of injected noise in the control on the state:

**Definition 11.** The noise transfer matrix is defined as $\Phi_i = A^i B$.

We have the following representation of the state

$$\mathbf{x}_{t+1} = A^{h+1} \mathbf{x}_{t-h} + \sum_{i=0}^{2h} \Psi_i(M^{1:h}) \mathbf{w}_{t-i} + \delta \sum_{i=0}^{h-1} \Phi_i \mathbf{n}_{t-i} \tag{9}$$

We are also interested in counterfactual state trajectories using non-stationary DAC policies. In particular, we have

**Definition 12.** The idealized state using policies $M^{1:h}$ is defined as

$$y_{t+1}(M^{1:h}) = \sum_{i=0}^{2h} \Psi_i(M^{1:h}) \mathbf{w}_{t-i} .$$

Similarly, the idealized cost is defined as

$$C_t(M^{1:h}) = c_t \left( y_t(M^{1:h}), \sum_{i=1}^{h} M_i^h \mathbf{w}_{t-i} \right) .$$

The univariate generalization of the idealized state and cost are

$$\tilde{y}_t(M) = y_t(\underbrace{M, M, \ldots M}_{h \text{ times}}), \quad \widetilde{C}_t(M) = C_t(\underbrace{M, M, \ldots M}_{h \text{ times}}).$$

We also define $F_t(\mathbf{u}_{1:h}) = c_t(\sum_{i=0}^{h-1} A^i (B\mathbf{u}_i + \mathbf{w}_{t-1-i}), \mathbf{u}_h)$ representing the instantaneous cost as a function of the last $h$ controls.

We note that

$$\widetilde{C}_t(M) = F_t(\sum_{i=0}^{h-1} M_i w_{t-i-h}, \sum_{i=0}^{h-1} M_i w_{t-i-h+1}, \cdots \sum_{i=0}^{h-1} M_i w_{t-i-1})$$

We now define a smoothed version of $F_t$, $F_{t,\delta}$ and a smoothed version of $\tilde{C}_t$, $\tilde{C}_{t,\delta}$ that uses $F_{t,\delta}$.

$$F_{t,\delta}(\mathbf{u}_{1:h}) = \mathbb{E}_{\mathbf{n}_{1:h} \sim \mathbb{S}_{d_u}} [F_t(\mathbf{u}_{1:h} + \delta \mathbf{n}_{1:h})]$$

$$\widetilde{C}_{t,\delta}(M) = F_{t,\delta}(\sum_{i=0}^{h-1} M_i w_{t-i-h}, \sum_{i=0}^{h-1} M_i w_{t-i-h+1} \cdots \sum_{i=0}^{h-1} M_i w_{t-i-1})$$

We also use the following notation for idealized costs fixing a realization of the exploration noise:

$$C_t(M|\mathbf{n}_{1:h}) = F_{t,\delta}(\sum_{i=0}^{h-1} M_i w_{t-i-h} + \mathbf{n}_1, \sum_{i=0}^{h-1} M_i w_{t-i-h+1} + \mathbf{n}_2, \ldots, \sum_{i=0}^{h-1} M_i w_{t-i-1} + \mathbf{n}_h) .$$

Since $\delta = o(1)$, the contribution to the state space is negligible, we can use bounds from Definition 5 of [Cassel and Koren, 2020]. In particular, we will use

$$h = \alpha^{-1} \log(2\kappa^3 T), \quad D_{x,u} = \max(10\alpha^{-1}\kappa^4 W(h\kappa + 1), 1) \tag{10}$$

*Proof of Theorem 9.* First, we state a bound on how large the states can get when modifying a DAC policy online.

**Lemma 13.** *Suppose controls are played according to $\mathbf{u}_t = \sum_{i=1}^h M_i^t \mathbf{w}_{t-i} + \delta\mathbf{n}_t$ where $\mathbf{n}_t \sim \mathbb{S}_{d_u}$ and $\delta = o(1)$, then*

1. *$\|\mathbf{x}_t\|, \|\mathbf{u}_t\| \leq D_{x,u}$ and $|c_t(\mathbf{x}_t, \mathbf{u}_t)| \leq CD_{x,u}^2$*

2. *Let $x_t^M, u_t^M$ correspond to the counterfactual trajectory, playing DAC with parameter $M$ for all time, then $|c_t(\mathbf{x}_t^M, \mathbf{u}_t^M) - \tilde{C}_t(M)| \leq \frac{GD_{x,u}^2}{T}$*

3. *$|c_t(\mathbf{x}_t, \mathbf{u}_t) - C_t(M^{t-h+1:t}|\mathbf{n}_{t-h+1:t})| \leq \frac{GD_{x,u}^2}{T}$*

The next lemma quantifies both the degree to which an idealized notion of cost tracks the true cost incurred for a DAC policy, and the resultant quality of gradient estimates thus obtained.

**Lemma 14.** *For all $t$, $C_{t,\delta}$ is convex and*

$$\|\nabla\widetilde{C}_{t,\delta}(M^t) - \mathbb{E}[\widehat{\nabla}_t]\| \leq \frac{2\eta d_u h^4 W^2 \kappa^3 \widehat{G} G D_{x,u}}{\delta} \,,$$

*and for all $M \in \mathcal{M}$,*

$$|\widetilde{C}_{t,\delta}(M) - \widetilde{C}_t(M)| \leq \delta h G D_{x,u} \kappa^3 \,.$$

We begin by observing that for any $t$ using Lemma 13.3 and the second part of Lemma 14, we have

$$|c_t(\mathbf{x}_t^M, \mathbf{u}_t^M) - \widetilde{C}_{t,\delta}(M)| \leq \frac{GD_{x,u}^2}{T} + \delta h G D_{x,u} \kappa^3.$$

A analogous result on the difference between true and idealized costs is stated below, but this time for the online algorithm itself which employs a changing sequence of DAC policies.

**Lemma 15.**
$$|c_t(\mathbf{x}_t, \mathbf{u}_t) - \widetilde{C}_t(M^t|\mathbf{n}_{t-h+1:t})| \leq \frac{GD_{x,u}^2}{T} + \eta GD_{x,u} W\kappa^3 h^2 \widehat{G}$$

Similarly, we have using Lemma 15 for any $t$ that

$$|c_t(\mathbf{x}_t, \mathbf{u}_t) - \widetilde{C}_{t,\delta}(M^t)| \leq \frac{GD_{x,u}^2}{T} + \eta GD_{x,u} W\kappa^3 h^2 \widehat{G} + \delta h G D_{x,u} \kappa^3.$$

Using this display, we decompose the regret of the algorithm as stated below.

$$\mathbb{E}\left[\sum_{t=1}^T c_t(\mathbf{x}_t, \mathbf{u}_t)\right] - \inf_{M \in \mathcal{M}} \sum_{t=1}^T c_t(\mathbf{x}_t^M, \mathbf{u}_t^M)$$

$$\leq \sum_{t=1}^T \widetilde{C}_{t,\delta}(M^t) - \inf_{M \in \mathcal{M}} \sum_{t=1}^T \widetilde{C}_{t,\delta}(M) + 2GD_{x,u}^2 + \eta GD_{x,u} W\kappa^3 h^2 \widehat{G} T + 2\delta h G D_{x,u} \kappa^3 T$$

Next, we use the following regret bound on an abstract implementation of online gradient descent with delayed updates, which we specialize subsequently to our setting.

**Lemma 16.** *Consider a delayed gradient update in Online Gradient Descent, executed as*

$$M^{t+1} = \Pi_{\mathcal{M}} \left[ M^t - \eta \widehat{\nabla}_{t-h} \right]$$

*where* $\|\mathbb{E}\nabla_t - \nabla\widetilde{C}_{t,\delta}(M^t)\| \leq \varepsilon$, $\|\widehat{\nabla}_t\| \leq \widehat{G}$, $D_{\mathcal{M}} = \max_{M \in \mathcal{M}} \|M\|$. *Additionally, if* $\max_{M \in \mathcal{M}} \|\nabla\widetilde{C}_{t,\delta}(M)\| \leq G_{\mathcal{M}}$, *then we have for any* $\eta > 0$ *that*

$$\mathbb{E}\left[ \sum_{t=1}^T \widetilde{C}_{t,\delta}(M^t) - \sum_{t=1}^T \widetilde{C}_{t,\delta}(M^*) \right] \leq 2\varepsilon D_{\mathcal{M}} T \sqrt{h d_{min}} + 2\eta h^2 d_{\min} \widehat{G}^2 T + \frac{2h d_{min} D_{\mathcal{M}}^2}{\eta}$$

Now, we invoke the regret upper bound from Lemma 16 to arrive at

$$\mathbb{E}\left[ \sum_{t=1}^T c_t(\mathbf{x}_t, \mathbf{u}_t) \right] - \inf_{M \in \mathcal{M}} \sum_{t=1}^T c_t(\mathbf{x}_t^M, \mathbf{u}_t^M)$$

$$\leq 2\varepsilon D_{\mathcal{M}} T \sqrt{h d_{min}} + 2\eta h^2 d_{\min} \widehat{G}^2 T + \frac{2h d_{min} D_{\mathcal{M}}^2}{\eta}$$

$$+ 2G D_{x,u}^2 + \eta G D_{x,u} W \kappa^3 h^2 \widehat{G} T + 2\delta h G D_{x,u} \kappa^3 T$$

Finally, we plug the value of $\widehat{G}$ from Lemma 17, and $\varepsilon$ from the first part of Lemma 14.

**Lemma 17.** *The stochastic gradients produced by Algorithm 3 satisfy the following bound*

$$\|\widehat{\nabla}_t\| \leq \widehat{G} := \frac{d_u h^2 W G D_{x,u}^2}{\delta}$$

As evident from the definition of $\mathcal{M}$, $D_{\mathcal{M}} = 2\sqrt{h}\kappa^4$. Setting $\eta = \sqrt{d_{min}/d_u} T^{-3/4}, \delta = \sqrt{d_u d_{min}} T^{-1/4}$ yields the result of $O(T^{3/4})$ regret for (possibly) non-smooth costs.

For the second part of the claim, we show an improved analogue of the second part of Lemma 14.

**Lemma 18.** *As long as $c_t$ is $L$-smooth, for all $M \in \mathcal{M}$, $|\widetilde{C}_{t,\delta}(M) - \widetilde{C}_t(M)| \leq 25 L \kappa^8 W^2 h^2 \delta^2 / \alpha$.*

Using this, in a manner similar to the derivation for non-smooth costs, we arrive at

$$\mathbb{E}\left[ \sum_{t=1}^T c_t(\mathbf{x}_t, \mathbf{u}_t) \right] - \inf_{M \in \mathcal{M}} \sum_{t=1}^T c_t(\mathbf{x}_t^M, \mathbf{u}_t^M)$$

$$\leq 2\varepsilon D_{\mathcal{M}} T \sqrt{h d_{min}} + 2\eta h^2 d_{\min} \widehat{G}^2 T + \frac{2h d_{min} D_{\mathcal{M}}^2}{\eta}$$

$$+ 2G D_{x,u}^2 + \eta G D_{x,u} W \kappa^3 h^2 \widehat{G} T + 50 L \kappa^8 W^2 h^2 \delta^2 T / \alpha$$

In this case, we set $\delta = (d_u d_{min})^{1/3} T^{-1/6}, \eta = d_{min}^{1/3}/(d_u^{2/3} T^{2/3})$ to arrive at the final bound as stated in the claim. $\qquad\square$

### E.4 Proof of Supporting Claims

*Proof of Lemma 13.* The properties follow from Lemma 6 in [Cassel and Koren, 2020], while using the fact that $\delta = o(1)$. $\qquad\square$

*Proof of Lemma 14.* Using the chain rule, we note that

$$\nabla\widetilde{C}_t(M) = \sum_{i=1}^h \left( \nabla_{\mathbf{u}_i} F_t(\mathbf{u}_{1:h})\big|_{\mathbf{u}_k = \sum_{j=0}^{h-1} M_j \mathbf{w}_{t-h+k-j-1} \forall k} \right) \otimes \mathbf{w}_{t-h+i-1:t-2h+i}$$

Now, we note that the smoothed function $F_{t,\delta}$ will satisfy

$$|F_{t,\delta}(\mathbf{u}_{1:h}) - F_t(\mathbf{u}_{1:h})| \leq \delta h G_F, \quad |\widetilde{C}_{t,\delta}(M) - \widetilde{C}_t(M)| \leq \delta h G_F$$

where $G_F$ is the Lipschitz constant of $F_t$ with respect to a single $u$. This follows, by a hybrid-like argument smoothing one argument at a time using standard smoothing results (see e.g. [Gradu et al., 2020] Fact 3.2).We note that $G_F$ can be bound by $GD_{x,u}\kappa^3$. Furthermore, this smoothing preserves convexity of $F_{t,\delta}$ and composition of a linear and convex function is convex, so $\widetilde{C}_{t,\delta}$ also remains convex.

The gradients of the smoothed function then has the following form due to Lemma 6.7 from Hazan et al. [2016].

$$\nabla_{\mathbf{u}_i} F_{t,\delta}(\mathbf{u}_{1:h}) = \mathbb{E}_{\mathbf{n}_{1:h}\sim\mathbb{S}_{d_u}}[\frac{d_u}{\delta}F_t(\mathbf{u}_{1:h}+\delta\mathbf{n}_{1:h})\mathbf{n}_i]$$

$$\nabla\widetilde{C}_{t,\delta}(M) = \mathbb{E}_{\mathbf{n}_{1:h}\sim\mathbb{S}_{d_u}}\left[\frac{d_u}{\delta}\sum_{i=1}^{h}\left(\underbrace{F_t(\mathbf{u}_{1:h}+\delta\mathbf{n}_{1:h})|_{\mathbf{u}_k=\sum_{j=0}^{h-1}M_j\mathbf{w}_{t-h+k-j-1}\forall k}}_{C_t(M|\mathbf{n}_{1:h})}\right)\mathbf{n}_i\otimes\mathbf{w}_{t-h+i-1:t-2h+i}\right] .$$

Rearranging, we have

$$\nabla\widetilde{C}_{t,\delta}(M^t) = \mathbb{E}_{\mathbf{n}_{t-h+1:t}\sim\mathbb{S}_{d_u}}\left[\frac{d_u C_t(M^t|\mathbf{n}_{t-h+1:t})}{\delta}\sum_{j=0}^{h-1}\mathbf{n}_{t-i}\otimes\mathbf{w}_{t-i-1:t-h-i}\right]$$

Now, to relate this to $\mathbb{E}_{\mathbf{n}_{1:h}\sim\mathbb{S}_{d_u}}[\widehat{\nabla}_t]$, we note in expression for $\nabla\widetilde{C}_{t,\delta}(M^t)$, we bound $c_t(\mathbf{x}_t, \mathbf{u}_t|\mathbf{n}_{t-h+1:t}) - C_t(M^t|\mathbf{n}_{t-h+1:t})$ via Lemma 15. Using bounds on $\mathbf{w}, \mathbf{n}$ along with this bound, we have

$$\|\nabla\widetilde{C}_{t,\delta}(M^t) - \mathbb{E}[\widehat{\nabla}_t]\| \leq \frac{d_u h^2 W}{\delta}\left(\frac{GD_{x,u}^2}{T}+\eta GD_{x,u}W\kappa^3 h^2\widehat{G}\right) \leq \frac{2\eta d_u h^4 W^2\kappa^3\widehat{G}GD_{x,u}}{\delta}$$

$\square$

*Proof of Lemma 15.* We start with triangle inequality

$$|c_t(\mathbf{x}_t,\mathbf{u}_t) - \widetilde{C}_t(M^t)| \leq |c_t(\mathbf{x}_t,\mathbf{u}_t) - \widetilde{C}_t(M^{t-h:t}|\mathbf{n}_{t-h+1:t})| + |\widetilde{C}_t(M^{t-h:t}|\mathbf{n}_{t-h+1:t}) - \widetilde{C}_t(M^t|\mathbf{n}_{t-h+1:t})|$$

The first term is handled via Lemma 13, so we only need to bound the second term.

$$\begin{aligned}
|\widetilde{C}_t(M^{t-h:t}) - \widetilde{C}_t(M^t)| &= |c_t(y_t(M^{t-h:t}), \sum_{i=1}^{h}M_i^t\mathbf{w}_{t-i}) - c_t(\tilde{y}_t(M^t), \sum_{i=1}^{h}M_i^t\mathbf{w}_{t-i})| \\
&\leq GD_{x,u}\|y_t(M^{t-h:t}) - \tilde{y}_t(M^t)\| \\
&= GD_{x,u}\|\sum_{i=1}^{2h}\Psi_i(M^{t-h:t})\mathbf{w}_{t-i} - \sum_{i=1}^{2h}\Psi_i(M^t\ldots M^t)\mathbf{w}_{t-i}\| \\
&= GD_{x,u}\|\sum_{i=1}^{2h}\Psi_i(M^{t-h:t}-(M^t\ldots M^t))\mathbf{w}_{t-i}\| \\
&\leq GD_{x,u}\|\sum_{i=1}^{2h}\Psi_i(M^{t-h:t}-(M^t\ldots M^t))\mathbf{w}_{t-i}\|
\end{aligned}$$

Now we note that each matrix $M_i^s$, only occurs in one term of the form $A^k B M_i^s \mathbf{w}_l$, so we can refine the bound above to

$$
\begin{aligned}
|\widetilde{C}_t(M^{t-h:t}) - \widetilde{C}_t(M^t)| &\leq GD_{x,u} W \kappa^3 (1-\alpha) \sum_{i=1}^{h} \|M^{t-i} - M^t\| \\
&\leq GD_{x,u} W \kappa^3 (1-\alpha) \sum_{i=1}^{h} \sum_{s=t-i}^{t} \|\eta \widehat{\nabla}_{s-h}\| \\
&\leq \eta GD_{x,u} W \kappa^3 h^2 \widehat{G} \, .
\end{aligned}
$$

Combining, we have

$$
|c_t(\mathbf{x}_t, \mathbf{u}_t) - \widetilde{C}_t(M^t)| \leq \frac{GD_{x,u}^2}{T} + \eta GD_{x,u} W \kappa^3 h^2 \widehat{G}
$$

$\square$

*Proof of Lemma 16.* Since $c_t$ is convex, so is $\widetilde{C}_{t,\delta}$. Using this fact and the observation that $M^t$ is independent of $\mathbf{n}_{t:t-h}$ used to construct $\widehat{\nabla}_t$ due to the delayed update of gradients, we have

$$
\begin{aligned}
&\widetilde{C}_{t,\delta}(M^t) - \widetilde{C}_{t,\delta}(M^*) \\
&\leq \langle \nabla \widetilde{C}_{t,\delta}(M^t), M^t - M^* \rangle \\
&\leq \mathbb{E}\langle \widehat{\nabla}_t, M^t - M^* \rangle + 2\varepsilon D_{\mathcal{M}} \sqrt{h d_{min}} \\
&\leq \mathbb{E}\langle \widehat{\nabla}_t, M^{t+h} - M^* \rangle + \|\widehat{\nabla}_t\|_F \|M^{t+h} - M^t\|_F + 2\varepsilon D_{\mathcal{M}} \sqrt{h d_{min}} \\
&\leq \mathbb{E}\langle \widehat{\nabla}_t, M^{t+h} - M^* \rangle + \eta h^2 \widehat{G}^2 d_{min} + 2\varepsilon D_{\mathcal{M}} \sqrt{h d_{min}}
\end{aligned}
$$

The gradient update can be rewritten as

$$
\begin{aligned}
\langle \widehat{\nabla}_t, M^{t+h} - M^* \rangle &\leq \frac{\|M^{t+h} - M^*\|_F^2 - \|M^{t+h} - \eta \widehat{\nabla}_t - M^*\|_F^2}{2\eta} + \frac{\eta \widehat{G}^2 h d_{min}}{2} \\
&\leq \frac{\|M^{t+h} - M^*\|_F^2 - \|M^{t+h+1} - M^*\|_F^2}{2\eta} + \frac{\eta \widehat{G}^2 h d_{min}}{2},
\end{aligned}
$$

where we use the fact that the projection operator is non-expansive, hence $M^{t+h+1}$ is closer in Euclidean distance to $M^*$ than $M^{t+h} - \eta \widehat{\nabla}_t$. Telescoping this, we have for any $M^*$ that

$$
\mathbb{E}\left[ \sum_{t=1}^{T} \widetilde{C}_{t,\delta}(M^t) - \sum_{t=1}^{T} \widetilde{C}_{t,\delta}(M^*) \right] \leq 2\varepsilon D_{\mathcal{M}} T \sqrt{h d_{min}} + 2\eta h^2 d_{min} \widehat{G}^2 T + \frac{2h d_{min} D_{\mathcal{M}}^2}{\eta}
$$

$\square$

*Proof of Lemma 17.* Plugging in line 6 from Algorithm 3 and using our bounds on the cost, we have

$$
\|\widehat{\nabla}_t\| \leq \frac{d_u GD_{x,u}^2}{\delta} \sum_{j=0}^{h-1} \|\mathbf{n}_{t-i}\| \|\mathbf{w}_{t-i-1:t-h-i}\| \leq \frac{d_u h^2 WGD_{x,u}^2}{\delta} \, .
$$

$\square$

*Proof of Lemma 18.* We first make note of the following characterization of idealized costs under smoothness due to Cassel and Koren [2020] (Lemma 7.2, therein).

**Lemma 19** (Cassel and Koren [2020]). *If $c_t$ is L-smooth, then the smoothed and non-smoothed variants of the idealized costs $\widetilde{C}_t, F_t, \widetilde{C}_{t,\delta}, F_{t,\delta}$ are L'-smooth, where $L' = 25L\kappa^8 W^2 h/\alpha$.*

Note that $\widetilde{C}_t, \widetilde{C}_{t,\delta}$ only differ in that the latter is a noise-smoothed version of the former. Let $\mathbf{u}_{1:h} = \left[ \sum_{i=0}^{h-1} M_i \mathbf{w}_{t-i-h}, \sum_{i=0}^{h-1} M_i \mathbf{w}_{t-i-h+1} \cdots \sum_{i=0}^{h-1} M_i \mathbf{w}_{t-i-1} \right]$. Using the fact the noise $\mathbf{n}_{1:h}$ is zero-mean and independent of $\mathbf{u}_{1:h}$, we create a second-order expansion using Taylor's theorem to conclude

$$
\begin{aligned}
&|\widetilde{C}_{t,\delta}(M) - \widetilde{C}_t(M)| \\
=&|\mathbb{E}_{\mathbf{n}_{1:h} \sim \mathbb{S}_{d_u}} F_t(\mathbf{u}_{1:h} + \delta \mathbf{n}_{1:h}) - F_t(\mathbf{u}_{1:h})| \\
\leq~& |\underbrace{\mathbb{E}_{\mathbf{n}_{1:h} \sim \mathbb{S}_{d_u}} \langle \nabla F_t(\mathbf{u}_{1:h}), \delta \mathbf{n}_{1:h} \rangle}_{=0}| + \frac{L'}{2} \|\delta \mathbf{n}_{1:h}\|_F^2 \\
\leq~& L' \delta^2 h.
\end{aligned}
$$

$\square$

