# OpenReview forum: "Online Nonstochastic Model-Free Reinforcement Learning"
_NeurIPS.cc/2023/Conference — NeurIPS 2023 poster_

### Official Review · Reviewer_ijfz · 2023-07-03

**Soundness:** 3 good
**Presentation:** 3 good
**Contribution:** 2 fair
**Rating:** 6
**Confidence:** 2

**Summary:**

This paper introduces an extension to model-free RL that has its roots in non-stochastic control. The main idea is to adjust a nominal control of a base policy (from e.g. a RL algorithm) by a control signal that can be computed from estimates of the process noise. As a main contribution, the authors present three distinct methods to compute those noise estimates.

**Strengths:**

- Control theory provides us with many ideas that can be exploited and possibly transferred to model-free RL algorithms. Thus, I think it is important to think about how such ideas can be utilized and adapted to make RL-algorithms more reliable, sample-efficient etc.
- The ideas are formally justified and the derivations appear to be mostly correct. For LDS, the authors provide convergence proofs of their method.
- The paper is well structured and well written. The contributions are clearly set out and the structure of the paper allows the reader to follow along easily.
- The authors provide example code for some of their experiments.

**Weaknesses:**

- The results suggest, that PD1 produces similar results to the baseline and that PD2 was only able to really outperform its baseline on one task (noisy hopper). The only PD-estimator that consistently beat its baseline was PD3, which uses a simulator to compute the disturbance signal. Now, the question arises, whether this is still a model-free algorithm or not, since we would need this model also during inference. The paper would benefit from more evidence, that at least PD2 provides stronger performance than the baseline. Furthermore, it would be interesting to see if you could use a learned model to compute the disturbance and how this would affect performance.
- In the definition of PD(1), in Line 169, the term on the right-hand side is called "gradient of a TD error.". I do not think that this is an accurate statement. Recall that the TD-Error is defined as
$$c(x_t, u_t) + \gamma V_\pi(x_{t+1}) - V_\pi(x_t).$$
From the definition of $Q$ and $V$ and $\mathbf{\hat w_t}$, we can derive
$$
\begin{align*}
&\gamma V_\pi(f(x_t, u) + w_t) - (Q_\pi(x_t, u) - c(x_t, u)) \\
&=\gamma V_\pi(\hat x_{t+1}) - (\underbrace{c(x_t, u) + \gamma \mathbb E_{x_{t+1}}[V_\pi(x_{t+1})]}_{=Q(x_t, u)} - c(x_t, u)) \\
&= \gamma (V_\pi(\hat x_{t+1}) -  \mathbb E_{x_{t+1}}[V_\pi(x_{t+1})]),
\end{align*}
$$
which I would not call a TD error (it does not measure the consistency of the value function over timesteps). It appears to me that it is more like a measure between the expected value of the next state and the value of the predicted next state.
- In the evaluation, the plots are kind of hard to read. I would recommend to increase the font size for the axis/legend, especially for Fig. 3.

**Questions:**

- TD-Error: (See above)
- Why does PD2 not improve the performance in noisy walker and only marginally in noisy ant?
- Line 173: Here the authors state that the gradient of $V_\pi$ can be efficiently estimated online. This needs more explanation in my opinion, since it is not clear how this is done and why we do not have this gradient term in equations (1) and (2).
- Code: You provided code for the LDS experiments. Why not also for the robotics environments?

**Limitations:**

The authors did not dicsuss the limitations of their approach.

---

> ### Author Rebuttal · Authors · 2023-08-08
>
> Thank you for your time and thoughtful review.  We address the main points below:
>
> Weaknesses:
>
> Our contribution is mostly conceptual and theoretical, thus we focused on methods with provable guarantees. Using a learned model to compute disturbances could be interesting, but we don’t see how to develop theory based upon such a learned model at this time, especially if it is nonlinear.
> PD3 indeed performs best from our methods, but having a simulator is common in many applications, thus we think our method can be potentially practical.
>
> Sorry for the confusion regarding PD1: what we mean in “TD error” is the update/error in expected SARSA (Sutton and Barto, 2018), which derives from TD error. We recognize that typically TD error is defined on state value functions (instead of state-action ones). We will clarify this in the final version. Briefly, our "TD error" is designed so that it is zero in expectation in absence of disturbances, thus deviations of it (on average) signal presence of unmodeled disturbances. A central result we point is that the expected value of its gradients can be used to recover the disturbances for linear systems.
>
> We are happy to increase font sizes as you suggest.
>
> Questions:
>
> PD2: We are not sure why PD2 does not improve performance of noisy walker and only marginally in noisy ant. This is an interesting question for future investigation.
>
>
> Gradient formula:  We can explain this in more depth in the paper. The gradient term is not in (1) and (2) because we use sampling to estimate this gradient.  We are using a single point zeroth order gradient estimator (See equation (1) in  https://arxiv.org/pdf/2006.06224.pdf ). In our case, with quadratic value functions there is no smoothing error as can be seen in the proof in Appendix D.1 of the supplement.
>
> Code: We could not release the code for the larger scale experiments because these were implemented and tested within a proprietary benchmarking suite and there are restrictions on release. We plan to implement the experiments in open source and release. More importantly, we are very happy to release the algorithm's code itself very soon.
>
> Limitations:
>
> We disagree that we did not discuss limitations of our methods. We discuss limitations of each signal in section 3.4 as is attested by Reviewer MnP5.  We agree that theory being restricted to linear dynamical systems is a limitation, though we never claim provable guarantees beyond this setting.
>
> Richard S Sutton and Andrew G Barto. Reinforcement learning: An introduction. MIT press, 2018.

---

> > ### Comment · Reviewer_ijfz · 2023-08-14
> >
> > I thank the authors for responding to concerns raised during my review. Furthermore, I want to apologize for overlooking the section where the limitations were discussed.
> >
> > Most of my concerns were addressed. However, my main point of criticism (weak performance of PD1&2, access to simulator for PD3) remains.
> >
> > > ... having a simulator is common in many applications, thus we think our method can be potentially practical.
> >
> > Yes, that is true. But isn't (model-free) RL a technique that is mainly applied when we do not have a model? Access to a model would allow to use planning algorithms, which do not suffer from typical model-free RL problems. I think that including PD3 in the paper is still insightful for comparison to PD1 and PD2. However, I would like to see more evidence that the general approach and the truly model-free disturbance signals are indeed more robust.

---

> > > ### Author Response · Authors · 2023-08-17
> > >
> > > Thank you for the quick response. We are glad we were able to address many of your questions.
> > >
> > > We would like to highlight three points in relation to concerns regarding the generality of our approach:
> > >
> > > 1. We had included (at the end of the rebuttal period) new figures in a PDF file on a common thread demonstrating faster training on sinusoidal and adversarial sign gradient disturbances for the pendulum environment using PD2. While we were limited in our ability to do massive experiments during this short rebuttal period, we hope that this adds to the body of evidence that underscores the generality of our approach.
> > >
> > > 2. While PD2 does not perform as well as PD3, it is fairly domain agnostic and does improve performance in settings of interest. For example, in all our experiments, we use the coordinates of the state for each auxiliary reward. This suggests simple, domain-agnostic auxiliary reward signals can deliver significant utility for PD2.
> > >
> > > 3. We agree that PD3 is limited in that it requires access to a potentially inaccurate simulator; inaccurate in that it doesn't capture the perturbations present in the real world . Even with access to an exact simulator, model-free methods are also used for planning (which one can query at arbitrary states, see e.g., [Guez et al.](http://proceedings.mlr.press/v97/guez19a/guez19a.pdf)), due to standard planning algorithms like iLQR, RRT, analytical policy gradient (a.k.a. direct backprop) exhibiting poor performance in presence of contact forces or high dimensionality (respectively). Similarly, in our setting where we permit rollouts in the real-world to deviate from the given simulator, this suggests that model-free methods (like DDPG) might offer better performance than classical planning approaches. In this context, our main contribution is developing model-free algorithms whose behavior degrades gracefully the larger the gap between the real world and the simulator.
> > >
> > >
> > > Finally, we hope that the points above address some of the concerns from the reviewer.

---

> > > > ### Comment · Reviewer_ijfz · 2023-08-21
> > > >
> > > > Thank you for your answer. I changed my evaluation score to weak accept.

---

### Official Review · Reviewer_4h3N · 2023-07-07

**Soundness:** 4 excellent
**Presentation:** 3 good
**Contribution:** 2 fair
**Rating:** 5
**Confidence:** 3

**Summary:**

This work introduces the notion of disturbance-based policies for model-free reinforcement learning, as opposed to traditional state-based policies. The disturbances capture unmodeled deviations from observed dynamics. In the model-free setting, since these disturbances are not known to the learner, the paper proposes three signals which can be used as pseudo-disturbances. These include the gradient of the TD error, the difference between auxiliary value functions of consecutive states, and the difference between the observed state and a simulated state. These three signals each require different assumptions and have their own advantages and disadvantages. These can recover the true perturbations up to linear transformations assuming time-invariant linear dynamical systems and a linear policy. A new algorithm, MF-GPC, is introduced which can adapt existing RL methods under this framework. Notably, this approach has sublinear regret bound under certain assumptions. Experiments on noisy versions of OpenAI gym environments demonstrate the effectiveness of the proposed method.

**Strengths:**

The paper presents a novel paradigm for model-free reinforcement learning based on unmodeled disturbances. There is an existing line of work on disturbance-based techniques which employ model-based control. However, this paper is the first work to extend this approach to the model-free setting, to the best of my knowledge.

The use of various signals as pseudo-disturbances is an original idea and the three proposed variants seem sound. The mathematical guarantees that these pseudo-disturbances are linear transformations of the true disturbance for linear systems, as well as the regret bound for MF-GPC, are valuable contributions and strengthen the quality of the paper. The concepts introduced in the paper are presented with sufficient clarity.

**Weaknesses:**

The practical applicability of the proposed framework raises some concerns. Among the three proposed pseudo-disturbances, PD3 requires an accurate simulator which is in most cases not available. PD2 can be applied to specific systems where there are additional signals available from the environment. PD1 seems to be the one most generally applicable to typical RL environments and dynamical systems, however it is unreliable and the empirical results demonstrate the ineffectiveness of this signal. See questions for detailed comments.

Another consideration is that the analysis considers linear dynamical systems and the regret bounds and the guarantees require further assumptions. While such constraints are typical when performing rigorous analysis, it does raise the question of how generally applicable this framework is to various other types of systems.

**Questions:**

1. For PD1, both the state-value and the action-value function are learned estimates. If so, calculating the pseudo-disturbances based on the error between two functions which are being learned online can be unreliable, especially in the early stages of training. This is acknowledged in the paper and the results do not offer much improvement. This is concering because among the three proposed variants, PD1 seems to be the most easily applicable to general RL setups.

2. For PD2, the experiments section mentions that the first few units of the last critic layer are used as V and Q. The reason for this choice is not clear to me, these units would not, in general, correspond to any cumulant function. An explanation from the authors would be helpful.

**Limitations:**

The paper does not address the limitations of the proposed framework. As mentioned in my review, the main limitation seems to be the usefulness of the pseudo-disturbances to generic environments and systems. In addition, the analysis is limited to linear dynamical systems, and it is not clear how this framework extends to other types of systems.

---

> ### Author Rebuttal · Authors · 2023-08-08
>
> Thank you for your time and thoughtful review.  We address the main points below:
>
> We note that PD2 is actually quite general, and does not require specific domain-engineered signals from the environment.  For example, in all our experiments, we use the coordinates of the state for each auxiliary reward. This suggests simple, domain-agnostic auxiliary reward signals can deliver significant utility for PD2. We acknowledge that PD3 generally requires a simulator, though it fits in the framework and can be useful when a simulator is available. We acknowledge PD1 did not perform well, despite this being our initial idea and the nicest one in theory. We suspect this is due to the PD1 estimator's high variance.  We leave improving this via variance reduction to future work
>
> Regret minimization in general online MDPs with adversarial dynamics is known to be computationally hard (Yadkori et al., 2013), so restricting the setting is necessary to make progress. We will add discussion of this in the paper. We believe our setting is still meaningful, and sheds light on general principles that apply to nonlinear settings.
>
> Questions:
>
> 1. Good question! While it’s true the Q,V functions are learned estimates, as long as the Bellman (backup) equation holds, this works.  Generally, one can increase the number of gradient-based update steps for the critic per episode to make sure this error is small. We will add a short discussion of this to the paper.
>
> 2. We were a bit imprecise when describing this in the paper and will clarify in the paper. We use the same architecture for critic networks but with a wider final layer (since the output is no longer a scalar) to produce values for each auxiliary reward. We are not reusing the representations of the original critic.
>
> Limitations:
>
> We disagree that we did not discuss limitations of our methods. We discuss limitations of each signal in section 3.4 as is attested by Reviewer MnP5.  We agree that theory being restricted to linear dynamical systems is a limitation, though we never claim provable guarantees beyond this setting.
>
> [1] Yasin Abbasi Yadkori, Peter L Bartlett, Varun Kanade, Yevgeny Seldin, and Csaba Szepesvári. Online learning in markov decision processes with adversarially chosen transition probability distributions. Advances in neural information processing systems, 26, 2013.

---

> > ### Author Response · Authors · 2023-08-17
> >
> > We thank the reviewer again for their time. In our rebuttal, we have (a) substantiated our claim of PD2’s generality, (b) listed the fundamental difficulty in extending any of our results to the non-linear setting, and (c) address the questions from the reviewer (e.g., regarding Q/V functions that are learned online).
> >
> > Are there any other concerns that we can address before the discussion period ends?

---

> > > ### Comment · Reviewer_4h3N · 2023-08-17
> > >
> > > I thank the authors for their response to my questions. I overlooked the limitations mentioned in the text relating to each of the PDs discussed - my apologies.
> > >
> > > My specific questions were addressed satisfactorily. However, the applicability of PD2 and PD3 still remain as concerns. I do believe this paper has merit and the various signals described are illustrative, but ultimately it still seems to apply to specific situations.
> > >
> > > >  For example, in all our experiments, we use the coordinates of the state for each auxiliary reward. This suggests simple, domain-agnostic auxiliary reward signals can deliver significant utility for PD2.
> > >
> > > The coordinates of the state are specific to the particular MDP and state space and are not domain-agnostic. If the states are available as images, then coordinates are not available. I still have the opinion that PD2 needs to be selected carefully to suit the environment.
> > >
> > > I also share the views of Reviewer ijfz on PD3. I believe a more convincing demonstration of the applicability and usefulness of the proposed PD-based approach can strengthen the position the paper significantly.

---

> > > > ### Author Response · Authors · 2023-08-21
> > > >
> > > > We are glad you are satisfied with our discussion of limitations and with the answers to your specific questions.
> > > >
> > > > Perhaps domain agnostic was an inexact phrasing of what we intended. What we really wanted to emphasize is that PD2 can be applied generally across domains via standard techniques without significant domain-specific effort. In this spirit, while we agree that our method might not perform well directly on image-based representations without further modification due to high dimensionality of the stated representation, we do think using coordinates of the neural net representation (e.g., activations of the pre-output layer of a neural net) of the state (as scalar rewards) is a natural way to adapt our algorithm in such settings. Using such representations for image-based domains is a standard practice in reinforcement learning;  for example, the use of conv nets in policies is often necessary for sample efficiency when dealing with images. We believe our methods are just as applicable as these standard adaptations made when switching domains in the RL literature. Using pre-trained embeddings can also be a viable alternative.
> > > >
> > > > We would like to point out that we had included (at the end of the rebuttal period) new figures in a PDF file on a common thread demonstrating faster training on sinusoidal and adversarial sign gradient disturbances for the pendulum environment using PD2. While we were limited in our ability to do massive experiments during this short rebuttal period, we hope that this adds to the body of evidence that underscores the generality of our approach.

---

### Official Review · Reviewer_MnP5 · 2023-07-07

**Soundness:** 3 good
**Presentation:** 4 excellent
**Contribution:** 3 good
**Rating:** 6
**Confidence:** 3

**Summary:**

This work considers model-free RL in the setting with additive disturbances in the environments' forward dynamics. In particular, it focuses on "disturbance-based policy" which adds a correction policy to the vanilla state-based policy. Because the disturbances are unknown in model-free RL, the correction policy instead conditions on "pseudo-disturbance" which are proxies of the actual disturbance. The paper proposes and analyzes 3 pseudo-disturbances that are feasible to compute in a model-free RL setting. The main algorithm trains the disturbance-based policy in a typical model-free RL training loop and is independent of the base RL algorithm. The authors provide theoretic guarantees of the method under linear settings. Empirical results show that the algorithm brings substantial improvement in a collection of noisy control tasks.

**Strengths:**

1. The problem setting is well-motivated and clearly formulated.
2. New concepts are nicely explained.
3. The different choices of pseudo-disturbance make sense intuitively and are well supported for their properties in the linear case.
4. Empirical results show significant benefits from the algorithm.

**Weaknesses:**

1. The authors mention potential adversarial disturbance but empirical results are shown in environments with uniform additive noise. The paper would be stronger with experiments containing adversarial disturbances.
2. The gym experiments are done with only one baseline method. Max-entropy RL algorithms like Soft Actor-critic might be more robust to noises in the environment. I would be more convinced if the method is compared with a few more state-of-the-art methods.

**Questions:**

This links back to weakness #2. I appreciate the fact that disturbance-based policies provide theoretic guarantees in linear settings. Does this formulation provide other benefits? How does the proposed method compare to other model-free RL methods?

**Limitations:**

The authors cover the limitation of each pseudo-disturbance. They are also upfront about PD1 not performing well in RL.

---

> ### Author Rebuttal · Authors · 2023-08-08
>
> Thank you for your time and thoughtful review.  We address the main points below:
>
> Weaknesses:
>
> We acknowledge that paper would be stronger with non-uniform disturbances. We will add detailed experiments for sinusoidal disturbances and adversarial sign gradient disturbances.
>
> Benchmarking against other more state-of-the-art methods like SAC would be useful, and we will add this. We will note that our algorithm is an augmentation or modification on top of a base algorithm, so we can apply our method on top of other baselines (SAC included) as well.  We wanted to show an improvement on the base algorithm with our experiments so this is a proof of concept for now.
> We have not completed this for the rebuttal because we did not have the time to run the appropriate hyperparameter sweeps, which are needed even for new baselines in our modified noisy environments.
>
> Questions:
>
> Beyond our theory for linear dynamical systems, we note that crucially MF-GPC policies are non-markovian (take more than immediate state into account), which could potentially help in non-markovian environments. This point is also highlighted by our theory for linear systems. In contrast, most of the RL algorithms are designed for markovian settings.

---

> > ### Comment · Reviewer_MnP5 · 2023-08-20
> >
> > Thank you for addressing my concerns and the new experiment results in particular. I agree that using some sign gradient attack in experiments provides a stronger narrative.
> >
> > PD3 could be useful in settings where people try to train RL agents on real robot hardware directly. In most cases, they have access to some imperfect simulator (imperfect because it is hard to recreate the exact task scenes in simulation, but the dynamics model of the robot itself is available). Experiments along this line might provide better justifications for this work.

---

> > > ### Author Response · Authors · 2023-08-21
> > >
> > > We are glad our additional experiment results were helpful to you. Indeed, we agree that PD3 has great potential in robotics settings involving an imperfect simulator, as you described. Our principal contribution here is the pseudo-disturbance framework and the principled design of pseudo-disturbance signals, backed by theoretical guarantees. Our experimental evaluations, in our view, serve as proof-of-concept exercises to demonstrate the potential for its real-world use; hence, we did not intend them to be exhaustive.

---

### Author Rebuttal · Authors · 2023-08-08

**Please see attached pdf file.**

Dear reviewers, we thank you for your time and effort in reviewing our paper!
The main weakness jointly pointed by the reviewers is lack of experimentation with more varied noise patterns, as opposed to stochastic noise.

We attach below in pdf format two experiments that show faster training for PD2 with:
1. sinusoidal disturbances
2. adversarial sign gradient attack disturbances (similar to [this](https://arxiv.org/pdf/1412.6572.pdf))

We will expand on these at the earliest chance of revision.

More detailed responses are addressed below to each reviewer separately.

Thanks again, we hope this will alleviate your concerns.

---

### Author Response · Authors · 2023-08-21
**Overlooked theory contribution (new SOTA regret bounds for bandit control)**

Having arrived at the end of the discussion period, we recount our contributions next. We identify three novel pseudo-disturbance signals, each occupying an incomparable region in the space of generality-performance tradeoff. We show that these can be used to train a disturbance-based policy via a new provably efficient gradient-based model-free algorithm for at-test-time adaption to unmodeled disturbances, and evaluate them empirically. We hope our responses have allayed the reviewers’ concerns, and we hope that they will consider raising their score.

We would also like to take the opportunity to highlight a **core contribution** that was perhaps not as conspicuous and was missed by some reviewers. As a testament to the strength of our careful algorithm design and analysis, as stated in Line 54 of the main paper, even for the well-studied linear setting, our sub-linear guarantee is unconditionally better than best-known ones (published at NeurIPS; Cassel and Koren, and Gradu et al) in terms of its dependence on the state dimension and applicability to high-dimensional systems.

Specifically, for bandit linear control, the best-known results offer a regret of $$O(\sqrt{d_xd_u d_{\min}} T^{0.75}),$$
whereas we obtain  $$O(\sqrt{d_u d_{\min}} T^{0.75}),$$
where $d_x$ is the state dimension, $d_u$ is the action dimension, and $d_{\min}=\min (d_u,d_x)$. This is an unconditional (quantitative) improvement over an established line of prior work.

Next, we comment on the qualitative significance of this improvement. For underactuated systems (where $d_u\leq d_x$ making these rank deficient systems harder to control), our results have no dependence on $d_x$ whatsoever; thus this algorithm is especially well suited to high dimensional state space settings. This improvement over prior work comes about due the the *correct* choice of the gradient estimator and not just an improved analysis, as we detail in Appendix E.2. (Additionally, as proved in E.2, such improvements in dimension dependence also apply to smooth losses, where the regret scales as $T^{2/3}$, unconditionally improving over an analogous result in Cassel and Koren).

This is further validated empirically in Figure 3, where our method outperforms the existing state-of-the-art bandit linear control algorithm (BPC) in the high dimensional experiment.

We could not adequately detail this in the main paper (we did on Line 54) due to space limitations, and due to the brunt of our focus being on the applicability and design of the pseudo-disturbance estimators. We will alter the presentation to emphasize these important contributions.

**Since this is a concrete and unconditional improvement over papers published at NeurIPS, we hope that reviewers can take into account this innovation of ours in their scores.**

---

### Decision · Program_Chairs · 2023-09-21

**Decision:**

Accept (poster)

**Comment:**

The paper addresses model-free RL when there are additive disturbances in the dynamic model, with the key idea to introduce a correction term that is derived from estimates of the process noise. All the reviewers are in favor of accepting the paper – the main criticism being empirical evaluation and lack of clarity/discussion on limitation of the method.

During the discussion phase the reviewers raised many questions that were satisfactorily answered by the authors. Moreover, the authors also shared additional experimental results with varied nose patterns – these experiments are in-line with the reviewers ask. Also, the authors point out an additional contribution (a better regret that shows effectiveness in higher dimensional state spaces) that they did not explicitly focus on in the original submission.

My recommendation is accept, and I encourage authors to both include the new experimental results as well as the novel theoretical insight. Additionally, a discussion of limitation of the method will be beneficial to the research community.